# Colloidal pathways of amorphous calcium carbonate formation lead to distinct water environments and conductivity

Maxim B. Gindele[1], Sanjay Vinod-Kumar [2], Johannes Rochau[1], Daniel Boemke[1], Eduard Groß[1], Venkata SubbaRao Redrouthu[2], Denis Gebauer [1] ✉ & Guinevere Mathies [2] ✉

CaCO$_3$ is the most abundant biomineral and a major constituent of incrustations arising from water hardness. Polycarboxylates play key roles in controlling mineralization. Herein, we present an analytical and spectroscopic study of polycarboxylate-stabilized amorphous CaCO$_3$ (ACC) and its formation via a dense liquid precursor phase (DLP). Polycarboxylates facilitate pronounced, kinetic bicarbonate entrapment in the DLP. Since bicarbonate is destabilized in the solid state, DLP dehydration towards solid ACC necessitates the formation of locally calcium deficient sites, thereby inhibiting nucleation. Magic-angle spinning (MAS) nuclear magnetic resonance (NMR) spectroscopy of poly-aspartate-stabilized ACC reveals the presence of two distinct environments. The first contains immobile calcium and carbonate ions and structural water molecules, undergoing restricted, anisotropic motion. In the second environment, water molecules undergo slow, but isotropic motion. Indeed, conductive atomic force microscopy (C-AFM) reveals that ACC conducts electrical current, strongly suggesting that the mobile environment pervades the bulk of ACC, with dissolved hydroxide ions constituting the charge carriers. We propose that the distinct environments arise from colloidally stabilized interfaces of DLP nanodroplets, consistent with the pre-nucleation cluster (PNC) pathway.

The mechanism of calcium carbonate (CaCO$_3$) formation has been of interest for many decades. This is not only due to its relevance in biomineralization processes[1–3], geological phenomena or industrial applications, e.g., as filler or pigment[4,5]. CaCO$_3$ also serves as a model system to investigate nucleation and crystallization phenomena in general. Multiple theories have been developed to describe the nucleation of mineral phases, ranging from classical nucleation theory to different, so-called nonclassical notions[6–9]. In particular, the prenucleation cluster (PNC) pathway, which describes phase separation involving stable solute ion clusters[10], has proven both its explanatory and predictive powers[11]. While the occurrence of liquid-like amorphous intermediates

as precursors of crystalline CaCO$_3$ was proposed two decades ago[12–14], it was only recently that the liquid-liquid phase separation (LLPS) process was explored in more detail[15]. Liquid-like intermediates have been observed in vitro[16,17] and in biominerals[18]. Avaro et al. presented the missing link between PNCs and these liquid-like intermediates and introduced a quantitative, nonclassical model to describe the LLPS phase separation process[19]. Further advancements include the observation that, at near-neutral pH values, HCO$_3^-$ ions are incorporated as a structural component in amorphous calcium carbonate (ACC) based on thermodynamic interactions with PNCs, which might play a significant role in biomineralization processes in seawaters[20–22].

[1]Institute of Inorganic Chemistry, Leibniz University Hannover, Callinstr. 9, 30167 Hannover, Germany. [2]Department of Chemistry, University of Konstanz, Universitätsstr. 10, 78464 Konstanz, Germany. ✉e-mail: gebauer@acc.uni-hannover.de; guinevere.mathies@uni-konstanz.de

Due to their high complexity and the numerous possibilities for interactions between the various chemical species, additive-controlled systems are still poorly understood compared to additive-free systems[23,24]. One of the most fascinating observations in additive-controlled $CaCO_3$ formation is the effective nucleation and crystallization inhibition by polycarboxylates, even at minute concentrations (low ppm range)[25–30] of the additives in aqueous solutions. Understanding the interactions of polycarboxylates with the various species occurring along the mineralization pathway is not only relevant in terms of their application as commercial scale inhibitors but also crucial to better understand biomineralization mechanisms—proteins associated with biomineral formation are often rich in Asp and Glu residues[31–37]. Polycarboxylates exhibit a strong stabilizing effect on both liquid-like and solid amorphous precursor phases[6,17,38]. Stabilization is usually attributed to the adsorption of the additive molecules on the formed colloidal minerals, preventing their dissolution[39,40], or to the adsorption of calcium ions in the solution[41]. Some studies hinted at the relevance of polycarboxylates in altering calcium (bi)carbonate ion association by modulating the pH value, resulting in the kinetic stabilization of a bicarbonate-rich mineral precursor[42]. However, even in systems employing precisely controlled higher pH levels and distinctly sub-stoichiometric amounts of polycarboxylates, a significant degree of inhibition can be detected[26]. Clearly, so far, there is no explanation for the effective inhibition of solid nucleation and crystallization by polycarboxylates.

Here, we show that significant amounts of bicarbonate ions are incorporated into the dense liquid phase (DLP) mineral precursor in the presence of polycarboxylates, while the extent of bicarbonate binding and the inhibition of nucleation of solid ACC from the DLP precursor, as observed in potentiometric titration experiments are proportional. We propose that bicarbonate incorporation into the DLP is due to kinetic entrapment upon aggregation and coalescence of polymer-stabilized DLP nanodroplets due to interface-bound bicarbonate, in agreement with the previously established LLPS mechanism via PNCs. Magic-angle spinning nuclear magnetic resonance (MAS NMR) spectroscopy detects only small amounts of bicarbonate incorporated in solid, polymer-stabilized ACC, however, bicarbonate fractions observed by thermogravimetric analysis coupled with mass spectrometry and infrared spectroscopy (TGA-MS-IR) are on par with those determined in the potentiometric titrations. This apparent contradiction can be reconciled by the formation of locally calcium-deficient sites in solid ACC that decompose via calcium bicarbonate upon heating. The induced local stoichiometric mismatches likely play a role in stabilizing solid ACC against crystallization. Further investigations by MAS NMR reveal the presence of two distinct water environments in ACC, with hydroxide ions residing in a mobile environment. Indeed, conductive atomic force microscopy (C-AFM) shows that the ACC exhibits conductivity, suggesting that the mobile environment forms a network through the bulk of ACC. These findings, too, can be rationalized in light of the PNC pathway.

## Results and discussion

### Potentiometric titrations show pH-dependent nucleation inhibition by polycarboxylates

Additive-controlled calcium carbonate formation can be explored utilizing a previously established potentiometric titration procedure[26]. Both poly(aspartic acid) (PAsp) and poly(glutamic acid) (PGlu) strongly inhibit nucleation, even at concentrations in the ppm range in solution (for details, see Section 3 in the Supplementary information (SI)), and in accordance with literature[26]. Inhibition of nucleation is quantified using a so-called scale factor, which compares the amount of calcium added until the nucleation of a solid takes place in the additive-containing experiment to the amount added in the additive-free reference experiment. A comparison of the pH-dependency of the scale factor for the polymers (Fig. 1a) shows a stronger inhibition in the

presence of PAsp compared to PGlu at all investigated pH values, and a strong pH-dependency of the scale factors is observed. While at pH 10.2, roughly 3 times stronger inhibition is detected for PAsp, the scale factor even exceeds a factor of 10 at pH 9.0.

### Polymers facilitate kinetic $HCO_3^{2-}$ entrapment within the dense liquid mineral phase

In search of mechanistic explanations for this strong inhibition and its pronounced pH-dependency, the pH titration data was quantitatively evaluated (Supplementary Fig. 1). In previous polymer-free experiments, calcium and carbonate ions are essentially bound at a 1:1 binding ratio in the prenucleation regime above pH 9.0[10,20]. In polymer-free experiments, carbonate ions and calcium ions are bound in PNCs, resulting in the formal removal of carbonate ions from the buffer equilibrium ($HCO_3^- \rightleftharpoons H^+ + CO_3^{2-}$). Protons are formed due to the regeneration of carbonate from bicarbonate ions upon re-equilibration at constant pH. In the titration experiment, these protons are then neutralized by automatic NaOH addition to keep the pH constant. Therefore, the extent of NaOH addition in the prenucleation regime can be tracked to assess the carbonate binding (Fig. 1b), confirming a calcium:carbonate ion binding ratio for polymer-free reference experiments of 1:1 (Fig. 1c). However, for experiments with polymers, NaOH addition is significantly lower than in the reference titration, while the amount of bound $Ca^{2+}$ is affected less. This observation raises the question of whether the calcium carbonate association is still 1:1 in the presence of the polymers. The difference between bound carbonate calculated from NaOH addition and calcium binding (Fig. 1c) can be explained by simultaneous binding of bicarbonate species in the prenucleation regime, while there is still $Ca^{2+}$–$CO_3^{2-}$ association: Simultaneous binding of bicarbonate and carbonate ions causes some carbonate ions to be invisible to the pH titration, due to the properties of the buffer equilibrium. In fact, at the pH value considered here (pH 9.8), the binding of 2.3 bicarbonate ions masks the binding of 1 carbonate ion in the presence of the polymers, allowing the quantification of the amount of bicarbonate binding from the recorded NaOH addition. The detailed calculations and reasoning for bicarbonate binding are discussed in Section 2 in the Supplementary information. Quantitative evaluation (Fig. 1d, black data points) of this effect suggests considerable bicarbonate binding, amounting to around 15% of all carbonate species in the prenucleation regime for PAsp and around 25% for PAA (polyacrylic acid). The order of the extent of bicarbonate binding (Reference → PGlu → PAsp → PAA, Fig. 1d, black data points) corresponds to that observed in the scale factors (Fig. 1d, orange bars).

The presence and relevance of bicarbonate binding within a $CaCO_3$ dense liquid phase (DLP) has been reported previously[22,42], however, the extent of bicarbonate binding could not be quantitatively assessed. Here, bicarbonate binding upon the formation of a DLP and its stabilization is in agreement with the stronger inhibition observed with decreasing pH values (Fig. 1a)—lower pH corresponds to a higher fraction of bicarbonate present in the solution. The bicarbonate binding seems to be facilitated by the polymer, which is known to have a stabilizing effect on the DLP. In any case, the observed bicarbonate binding is at odds with previously established binding constants[20,43], indicating that the effect is kinetic in nature. We will return to discuss the mechanistic basis for this later and first explore whether or not any polycarboxylate-facilitated effects are transferred into solid ACC.

### Bicarbonate entrapment in liquid precursors leads to locally calcium- and carbonate-deficient sites in polymer-stabilized, solid ACC

To gain further insights into polymer-stabilized solid ACC that forms via dehydration of the precursor DLP[11,19], solid ACC samples were isolated by quenching the titration at pH 9.8 shortly before reaching the maximum in free ion products (see Methods section) by pouring the

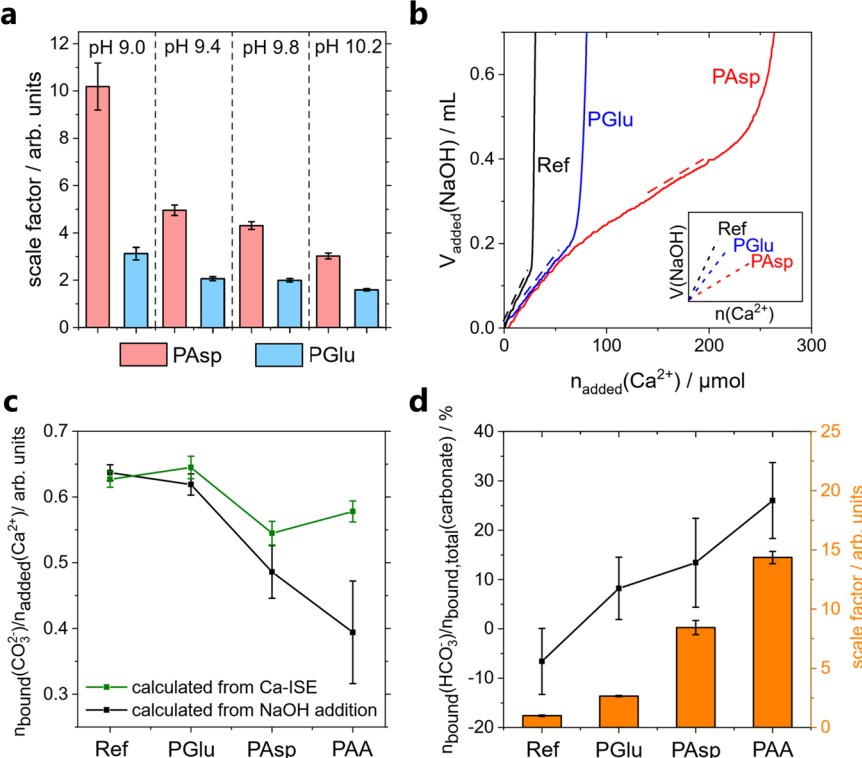

**Fig. 1 | Potentiometric titration experiments in the presence of polymer additives. a** pH dependency of the scale factor (crystallization time relative to reference) for experiments containing 10 mg/L PAsp (red) and PGlu (blue). **b** Amount of NaOH added in titration experiments at pH 9.0 containing 10 mg/L polymer. The dotted lines visualize the slope of added NaOH in the prenucleation regime close to nucleation. In experiments with polymer, the extent of NaOH addition was significantly lower (see inset for easier comparison of slopes). **c** Development of molar amount of bound $CO_3^{2-}$ per added $Ca^{2+}$ in the prenucleation regime for experiments

containing 0.1 g/L (PAsp and PGlu) or 0.01 g/L (PAA) polymer at pH 9.8. Values are calculated from the Ca-ISE (green) by assuming a 1:1 $Ca^{2+}$:$CO_3^{2-}$ binding ratio and from the NaOH addition (black) by calculating the amount of bound $CO_3^{2-}$ from changes in buffer equilibria. **d** From the values shown in (**c**), the proportion of bicarbonate binding relative to the total amount of bound carbonate species ($CO_3^{2-}$ + $HCO_3^-$) can be calculated (black). Error bars represent ±1 − $\sigma$-standard deviation. The scale factors for each experiment are also shown (orange bars). The calculations are described in detail in Section 2 in the Supplementary Information.

reaction solution into a large excess of ethanol[44]. The quenching procedure causes rapid dehydration of the DLP, yielding the respective polymer-stabilized solid ACC sample (PAsp_ACC), while the large excess of ethanol also prevents crystallization of the ACC. As shown earlier[44], quenching of additive-free titration experiments before the nucleation event leads to ACC nanoparticles with several tens of nm in size (see also Section 3.2 in the Supplementary information). The isolated sample was then characterized to explore if the above-discussed bicarbonate binding (i.e., within the DLP precursor in the presence of the ppm amounts of polymers in the solution) can also be observed in solid ACC.

Samples of PAsp-stabilized ACC were investigated by magic-angle spinning (MAS) NMR spectroscopy at 9.4 T ($^1$H Larmor frequency of 400 MHz, experimental procedures are described in detail in the Methods section). Figure 2a shows $^{13}$C spectra after direct excitation (black curve) and after a cross-polarization transfer from $^1$H nuclei (blue and red curves). With $^1$H–$^{13}$C cross-polarization spectra of PAsp in hand (Supplementary Fig. 2), the assignment of the observed resonances is readily performed. The prominent peak at 168.9 ppm (line width 3.3 ppm) is due to the carbonate of ACC[45–47]. The low-field shoulder and the peaks in the aliphatic region are from the $^{13}$C nuclei of Asp (the peak at 69.2 ppm is a spinning sideband of the carbonate signal). The small foot around 162 ppm is due to bicarbonate. $^1$H–$^{13}$C correlation spectra obtained with frequency-switched Lee-Goldburg homonuclear decoupling along the $^1$H dimension confirm these assignments (Supplementary Fig. 3). Specifically, cross-peaks at 4.5 and 161.9 ppm and at 11.0 and 169.3 ppm in Supplementary Fig. 3 indicate dipolar couplings between the $^1$Hs

of the water of ACC and the $^{13}$C of bicarbonate and between the $^1$H of bicarbonate and the $^{13}$C of the carbonate of ACC, respectively, which confirms the presence of bicarbonate and its structural incorporation[20] into ACC.

Comparison of the direct-excitation spectrum (Fig. 2a, black curve) to the cross-polarization spectra (Fig. 2a, blue and red curves) reveals that the polarization transfer to the carbonate of ACC is inefficient. The effect is less pronounced at −25 °C and the shape of the carbonate signal is not altered[44–46], making restricted motion a likely cause. This emphasizes that $^1$H–$^{13}$C cross-polarization and correlation spectra, although crucial for the identification of $^{13}$C-species, should not be used for quantitative analysis. For this purpose, only direct excitation spectra are suitable. After correction for differences in longitudinal relaxation rates, the direct excitation spectrum in Fig. 2a (black curve) indicates a PAsp content of 13% (m/m). In addition, analysis of the direct excitation spectrum in Supplementary Fig. 4 (taken of a sample prepared with 100% $^{13}$C-carbonate) indicates a bicarbonate/carbonate ratio of 1–2%. This ratio is drastically lower than the ratio found from potentiometric titration for the DLP precursor above (>10%), suggesting that the kinetically entrapped bicarbonate content is not transferred into solid ACC upon quenching. In fact, the bicarbonate content and the NMR spectral characteristics are reminiscent of additive-free ACC explored in previous work[20], indicating that this might be the maximum amount of bicarbonate that can be accommodated in the solid-state ACC structure, with and without additives. However, as will be discussed in detail below, we propose that bound bicarbonate reacts with other species upon dehydration and solidification of the DLP into solid ACC.

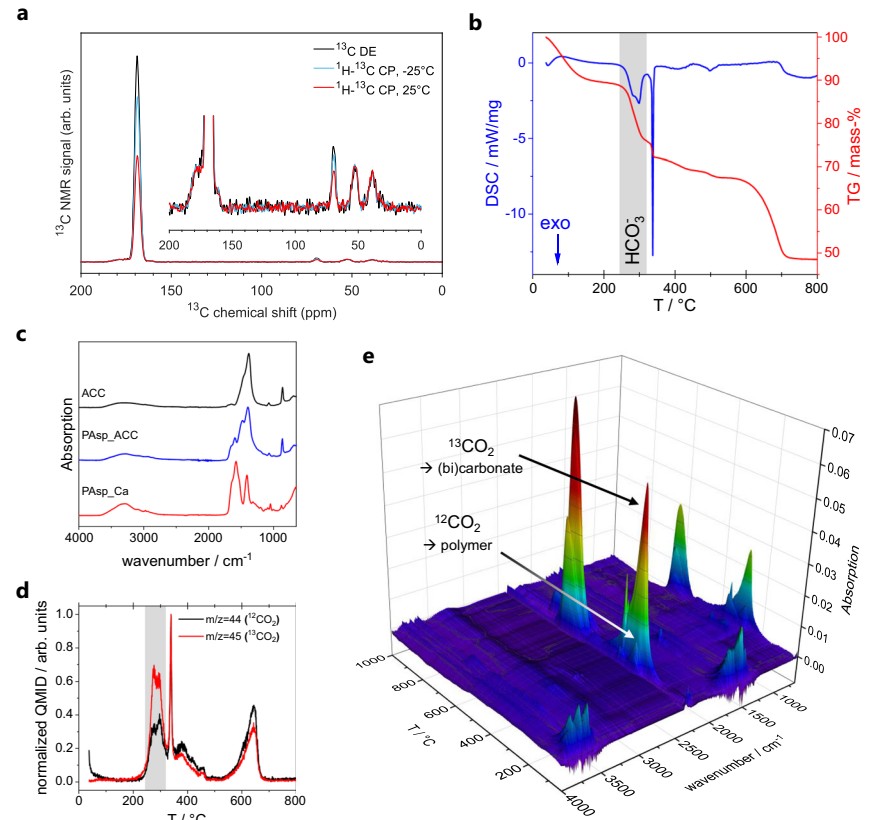

**Fig. 2 | Characterization of isolated polymer stabilized ACC.** The sample was isolated from a titration experiment using 0.1 g/L PAsp at pH 9.8 by quenching the solution in ethanol (see Methods section). **a** $^{13}C$ direct excitation (DE) and $^1H$–$^{13}C$ cross-polarization (CP) spectra of 10% $^{13}C$-carbonate ACC stabilized by PAsp (PAsp_disACC) at a spinning frequency of 10 kHz. The spectra are scaled at the $C_\alpha$-peak of PAsp. **b** TGA (red) and DSC (blue) analysis. The exothermic decomposition of the bicarbonate species is highlighted in grey. **c** ATR-FTIR spectra of polymer-stabilized ACC sample, showing significant amounts of polymer incorporation. Pure ACC and PAsp calcium salt (PAsp_Ca) are shown as references (detailed FTIR

spectra are shown in Supplementary Fig. 6). **d** Normalized QMID for TGA-MS measurement on the PAsp_ACC sample using $^{13}C$ enriched carbonates in the titrations. Due to the natural abundance of carbonate distribution in the polymer, released gases from polymer ($^{12}CO_2$; m/z = 44, black) and from mineral ($^{13}CO_2$; m/z = 45, red) can be distinguished, showing significant amounts of mineral decomposition below 300 °C (highlighted in grey). **e** TGA-IR analysis of the $^{13}C$ carbonate enriched PAsp_ACC sample confirms the strong $^{13}CO_2$ release from (bi)carbonate species at around 300 °C.

To explore the fate of the DLP-bound bicarbonate upon transferring it into the solid state further and independently assess the contents of organics in the ACC, we conducted thermogravimetric analysis (TGA). Decarbonization of ACC takes place at 600–800 °C, while the polymer exhibits a sharp (exothermic) decomposition at 300–400 °C, as visible from the pure ACC and PAsp reference TGA measurements (Supplementary Fig. 5). The PAsp-stabilized ACC sample presents the characteristic ACC decarbonization at high temperature, as well as a sharp exothermic feature due to polymer decomposition (Fig. 2b). The incorporation of significant amounts of polymer in the ACC sample is confirmed by ATR-FTIR investigations, as polymer vibrational bands are clearly visible in the PAsp_ACC sample (Fig. 2c).

TGA also reveals an exothermic decomposition process occurring prior to polymer decomposition (Fig. 2b, grey area). Analysis of the gas released during this step with mass spectrometry (TGA-MS) reveals $H_2O$ and $CO_2$ (Supplementary Fig. 7). This is expected for temperature-induced decomposition of bicarbonate ions via formation of the respective carbonate salt and release of $H_2O$ and $CO_2$ (e.g., for sodium bicarbonate: $2\ NaHCO_3 \rightarrow Na_2CO_3 + H_2O \uparrow + CO_2 \uparrow$). Polymer decomposition takes place, at least to some extent, at this same temperature (Supplementary Fig. 7b), possibly triggering the decomposition of a bicarbonate species. On the other hand, polymer decomposition occurs in a similar temperature range and also results in the release of $CO_2$ and $H_2O$. Indeed, PAsp calcium salt shows vastly different decomposition characteristics when compared to PAsp sodium salt

(Supplementary Figs. 7 and 8), revealing additional decomposition stages due to the subsequent formation and decomposition of $CaCO_3$[48].

To elucidate whether the observed process is due to mineral or polymer decomposition, $^{13}C$ enriched carbonates were employed in the synthesis of the polymer-stabilized ACC, allowing us to distinguish the $CO_2$ release arising from polymer decomposition ($^{12}CO_2$ release) from the decomposition of mineral (bi)carbonate species ($^{13}CO_2$ release). TGA-MS of the labeled sample clearly reveals a strong release of $^{13}CO_2$ prior to polymer decomposition (Fig. 2d, grey area), indicating that significant (bi)carbonate decomposition takes place at this temperature. We also detect $^{12}CO_2$ in this temperature region, showing that some polymer decomposition takes place simultaneously, confirming the TGA-MS results discussed above (Supplementary Fig. 7b). The release of $^{12}CO_2$ at higher temperatures is then caused by the decomposition of polymer carboxylate groups that are bound to $Ca^{2+}$ in the ACC sample, forming $CaCO_3$ as an intermediate upon their decomposition (Supplementary Fig. 7d), in accord with literature[48]. Additional TGA-IR characterization of the decomposition gases confirms the strong release of $^{13}CO_2$ around 300 °C (Fig. 2e). The $^{13}CO_2$ release in this temperature range was only detected for the polymer-stabilized ACC sample synthesized by titration, and neither pure $^{13}C$ enriched ACC, pure polymer, nor a mixture of independently isolated polymer and $^{13}C$ enriched ACC did exhibit this characteristic release (Supplementary Fig. 9).

Weight loss observed by TGA-MS-IR (-13%, corresponding to roughly 27% of bicarbonate present in the ACC, see Supplementary Fig. 9e) agrees with the amount of bicarbonate kinetically entrapped within the DLP as determined in the titration experiments (Fig. 1d). Yet, direct investigation of intact ACC nanoparticles by MAS NMR indicates a much lower bicarbonate content. To accommodate this apparent discrepancy, we propose the following reactive processes.

(i) Upon dehydration and solidification, DLP-entrapped bicarbonate ions react with $OH^-$ ions that are abundant in the dense and lean aqueous phases at basic conditions. While the $OH^-$/bicarbonate population is equilibrated in the liquid state, without hydration water, the polarizing power of $Ca^{2+}$ destabilizes the bicarbonate ion in the solid state. Consequently, no solid calcium bicarbonates are known to occur at ambient conditions, and 1–5% bicarbonate may be the maximum amount that can be accommodated within solid ACC[20]. Hence, upon dehydration and solidification of the DLP, the following process occurs: $DLP•HCO_3^- + OH^- \rightarrow ACC•CO_3^{2-}•H_2O$ (note that ACC usually contains an excess of around 1 mole of water per mole of calcium carbonate).

(ii) The kinetic entrapment of bicarbonate ions causes the DLP to be locally calcium-ion-deficient with respect to carbonate, as two bicarbonates are expected to balance the charge of one calcium ion within the macroscopically neutral DLP. This imbalance is transferred into solid ACC by process (i). Locally, structural water molecules stabilize these sites, which we label as $Ca^{(-)}$. Upon heating in TGA and removal of stabilizing water molecules, carbonate ions close to calcium-deficient sites then decompose according to the following process: $Ca^{(-)}CO_3•H_2O \rightarrow [Ca^{(-)}HCO_3]^+ + OH^- \rightarrow Ca^{(-)}(OH)_2 + CO_2\uparrow$.

Indeed, we can detect a corresponding calcium hydroxide fraction in XRD measurements of the samples after TGA analysis (Supplementary Fig. 10). The above processes lead to calcium carbonate decomposition via calcium bicarbonate at correspondingly low temperatures in TGA. Quantitatively, this corresponds to the extent of kinetic bicarbonate entrapment in the DLP, even though not being present as bicarbonate in solid ACC at ambient conditions. Rather, the polycarboxylate-mediated bicarbonate entrapment in the DLP leads to locally calcium-deficient solid ACC, which can markedly contribute to

the stabilization against nucleation of a solid. Thus, we observe a direct correlation between these effects, as discussed above. This, however, begs the question of how the charges are balanced within the ACC phase globally, as phases are certainly macroscopically neutral. Sodium ions do not seem to play a major role: The exothermic decomposition of bicarbonate species in polymer-stabilized ACC (Fig. 2b) differs fundamentally from that of $NaHCO_3$, which exhibits endothermic decomposition (Supplementary Fig. 5d). In addition, no sodium-containing phase was detected in the XRD spectra of the product isolated after TGA (Supplementary Fig. 10), further confirming that no significant $NaHCO_3$ coprecipitation took place upon the quenching and isolation of the ACC. It is possible that some sodium ions are incorporated within the ACC for local charge balancing, but it rather appears that negative charges at calcium-deficient sites may be globally balanced by excess calcium binding within the phase in other environments, likely in the proximity of the incorporated polymer. In any case, all of the above implies that the water sites in the polymer-stabilized ACC can play a crucial role, and these were further studied by MAS NMR.

## MAS NMR reveals two distinct water environments in polymer-stabilized ACC

The directly-detected (DD) [1]H NMR spectrum of ACC stabilized by poly-aspartate at a spinning frequency of 10 kHz is shown in Fig. 3a (black curve, analogous spectra at spinning frequencies of 5 and 0 kHz are shown in Supplementary Fig. 11). A pattern of spinning sidebands is observed as well as a strong central peak with a broad base, which appears to push this pattern up. This agrees with observations by others for ACC of both synthetic and biological origin[45,46,49–51]. A complete interpretation of the [1]H NMR spectra of ACC has, however, not been given.

By means of 2-dimensional WISE (wideline-separation) experiments (Supplementary Figs. 14 and 15)[52], we obtained [1]H spectra of ACC, which are detected after a transfer of magnetization, via cross-polarization, to the [13]C nucleus of carbonate (purple curves in Fig. 3a, Supplementary Fig. 11). Michel et al. already noted that for ACC these indirectly detected [1]H spectra differ from the directly detected [1]H spectra[45]. The strong central peak is absent and a Pake pattern, with spinning sidebands, due to the dipolar coupling between the two [1]Hs of water is discernable[53]. This is characteristic of structural water in minerals with a low hydrogen density[54].

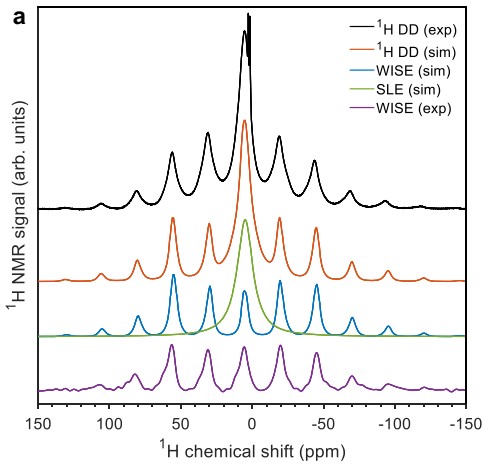

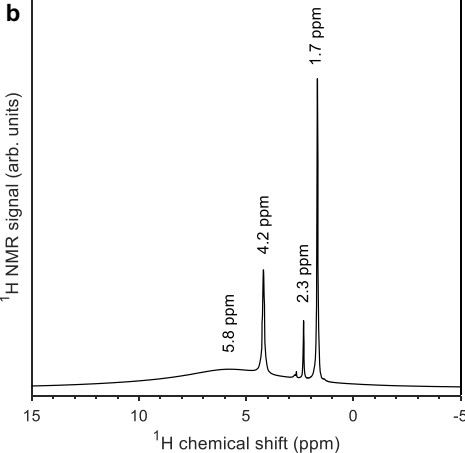

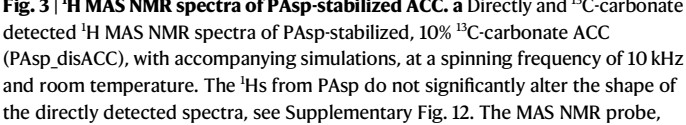

**Fig. 3 | [1]H MAS NMR spectra of PAsp-stabilized ACC. a** Directly and [13]C-carbonate detected [1]H MAS NMR spectra of PAsp-stabilized, 10% [13]C-carbonate ACC (PAsp_disACC), with accompanying simulations, at a spinning frequency of 10 kHz and room temperature. The [1]Hs from PAsp do not significantly alter the shape of the directly detected spectra, see Supplementary Fig. 12. The MAS NMR probe, however, gives rise to a broad [1]H NMR background signal, which has been removed from the directly detected [1]H spectrum (black) following a procedure outlined in the Supplementary information (Supplementary Fig. 13). **b** Central region of the directly detected [1]H NMR spectrum of 100% [13]C-carbonate ACC stabilized by PAsp (PAsp_disACC), at a spinning frequency of 10 kHz and room temperature.

To assist in the interpretation of the $^1$H spectra of ACC, we performed numerical simulations using the magnetic resonance simulation library Spinach[55]. We first simulated the indirectly detected $^1$H spectra, taking the magnetic properties of monohydrocalcite (MHC) as a starting point. The crystal structure of MHC is known from X-ray diffraction and has also been investigated by quantum chemical methods[56–59]. Simulations based on the calculated magnetic properties fit the experimental MAS NMR $^1$H spectra of MHC remarkably well (Supplementary Fig. 16). At room temperature, water molecules in many crystal hydrates undergo twofold rotations about the bisector axis (180° flips)[60]. If such flips are included in the simulation, the set of chemical shift anisotropies and dipolar couplings of MHC also produces a good simulation of the indirectly detected $^1$H spectra of ACC (Fig. 3a, Supplementary Fig. 11, blue curves). This strongly suggests that the $^{13}$C-detected $^1$H spectra arise from structural water molecules embedded in a rigid (but amorphous) environment of calcium carbonate, which allows only restricted, anisotropic motion.

Next, we turned to the simulation of the directly detected $^1$H spectra. The lack of structure (apart from a set of narrow lines, vide infra) and the width of the strong central peak (5–6 kHz) point to slow isotropic motion. To investigate this, we simulated the $^1$H MAS NMR spectra of a water molecule undergoing rotational diffusion with varying correlation times using the stochastic Liouville equation (SLE), see Supplementary Fig. 17[61,62]. For a correlation time of $1 \times 10^{-6}$ s, the simulated line width matches the width of the central peak in the experiments (green curves in Fig. 3a, Supplementary Fig. 11). Taking the sum of the simulations of the indirectly detected $^1$H spectra and the simulations of slow isotropic motion, here with equal weights, produces the red simulations in Fig. 3a, Supplementary Fig. 11, which reproduce all features of the experimental spectra (black curves).

The $^1$H NMR spectra and the numerical simulations thus indicate the presence of a second, mobile environment in ACC, which allows water molecules to undergo isotropic motion. The correlation time of this motion is considerably longer than for bulk water, likely due to spatial confinement. As the temperature is lowered from 25 to −25 °C (Supplementary Fig. 18), the isotropic motion slows down further, causing the central peak to come down while the broad base comes up.

We previously noted that the magnetization transfer by cross-polarization is inefficient for the $^{13}$C nuclei of carbonate in ACC (Fig. 2a). From the WISE experiments, we now know that the structural water molecules, which undergo restricted, anisotropic motion, are the source of this magnetization. Numerical simulations in Supplementary Fig. 19 show that the 180° flips we invoked to simulate the indirectly detected $^1$H spectra of ACC can also account for an inefficient transfer in a cross-polarization experiment. $^1$H–$^{13}$C cross polarization spectra recorded at a slow spinning frequency (2 kHz, Supplementary Fig. 20) reveal the chemical shift anisotropy of the $^{13}$C nuclei of carbonate in ACC. The observed pattern of spinning sidebands does not change as the temperature is decreased to −25 °C and is the same as recently observed for carbonate in a frozen carbonate solution (pH 7) at 100 K[63]. Thus, carbonate itself does not undergo motion or only on very slow time scales. Furthermore, directly detected $^{43}$Ca MAS NMR spectra show only broad, featureless resonances, hence giving no indication of mobility of calcium ions[64]. These observations together confirm that the rigid environment consists of amorphous calcium carbonate with embedded structural water molecules, which undergo restricted, anisotropic motion, and 1–2% of bicarbonate.

Figure 3b shows the central region of the directly detected $^1$H NMR spectrum of ACC in detail. A set of narrow lines (widths in the range of 15–50 Hz) is superimposed on the strong central peak (width 5–6 kHz). We assign these lines to hydroxide ions in rapid exchange with the water molecules that make up the second mobile environment[46]. Possibly the different observed chemical shifts relate to local variations in basicity. The presence of hydroxide ions within the mobile water environment can make ACC conductive. We explored this experimentally.

## ACC conducts electrical current

Established methods to measure (single) particle conductivity require connecting the particles to electrodes[65–67]. However, these methods are either not suitable for small sizes of particles (in our case, below 100 nm particle size) or only one particle can be measured at a time, resulting in poor statistics. We therefore developed a straightforward method for the qualitative assessment of the conductivities of individual particles based on conductive atomic force microscopy (C-AFM). The method utilizes measurements in C-AFM spectroscopy mode, i.e., recording an $I/V$ diagram on individual particles. Therefore, it is important to record the distance of the tip to the substrate during the spectroscopy measurement to confirm that the tip is indeed in contact with a particle. The slope of the $I/V$ diagram can then be compared to reference samples (conductive and non-conductive particles), to gain qualitative Information on conductivity. The method allows measurements of a large number of particles starting from a size of a few nm. Detailed descriptions of the method and data evaluation are provided in the Methods section and in Section 4 of the Supplementary information, respectively. To validate the method, gold nanoparticles (as conductive reference particles) and vaterite nanoparticles (as non-conductive reference particles) were synthesized and investigated, showing that the difference in conductivity can reliably be determined above a distance of ca. 20 nm (Supplementary Fig. 21). The C-AFM experiments were performed in a glovebox in water-free atmosphere and samples were dried in vacuum prior to measurement to eliminate potential effects of surface-adsorbed water on the measured conductivities.

Measurement of individual, sub-100 nm polymer-stabilized ACC particles showed conductivity for a measuring height below 10 nm (Fig. 4a, b), while above 10 nm, the conductivity (i.e., the slope of the $I/V$ diagram) gradually decreased to 0. This is because the increasing Z-height is caused by several particles lying on top of each other rather than by particles with increasing size, as visible by the AFM height images (Supplementary Fig. 22), and conductivity is likely poor across particles that are in loose contact with each other. Due to limitations of the C-AFM measurements for Z-heights below 20 nm owing to leaking voltage, we thus cannot provide unambiguous evidence for the conductivity of individual ACC particles at this point (see Supplementary Fig. 23 and related discussions). However, we also detected larger structures in polymer-stabilized ACC samples (Fig. 4c, d). These structures are likely dried residues of larger polymer-stabilized DLP species, which putatively formed in solution due to aggregation and partial coalescence of nm-sized ACC precursors before quenching. These structures have a smooth surface and are different from loosely aggregated individual particles, appearing as a continuous phase. All of these structures exhibit electrical conductivity across length scales that are orders of magnitude larger than single ACC particles (Fig. 4d), with the largest distance being almost 1 μm (Supplementary Fig. 24). ACC synthesized in the absence of polymer exhibits the same behavior; loose agglomerates of particles are not conductive (Supplementary Fig. 23h, i), while larger, continuous structures are (Fig. 4e, f).

## Interpretation of the two environments in ACC

The detection of two distinct water environments is in line with the proposed formation mechanisms of the ACC particles[11] (see Fig. 5 for an illustration). The key step of this process is the aggregation of phase-separated DLP nanodroplets, which is driven by the reduction of interfacial free energy. The liquid nanodroplets are colloidally stabilized by polycarboxylate molecules binding to calcium ions on their surface, leading to the formation of large, loose agglomerates. These can be visualized by cryogenic electron microscopy and have been erroneously interpreted as solid/liquid interfaces previously[11,68]. Other

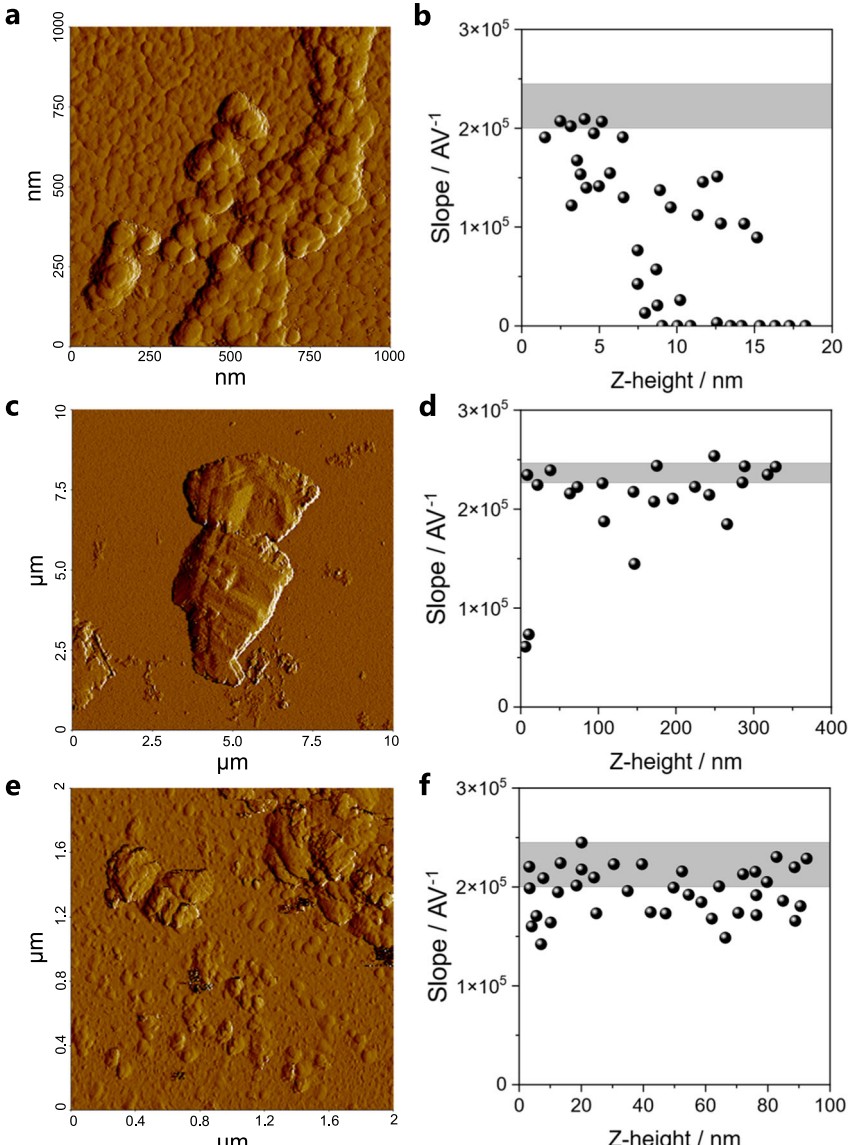

**Fig. 4 | Conductivity measurements of ACC samples using C-AFM. a** AFM amplitude map recorded in non-contact mode (NCM) during the investigation of a polymer stabilized ACC sample (PAsp_ACC). Individual ACC nanoparticles are visible. **b** Results of C-AFM spectroscopy measurements of polymer-stabilized ACC of the area as shown in (**a**). The slope of the I/V diagram close to the origin is shown versus the Z-height of the respective measurement point. The grey part shows the standard deviation of 100 measurements on the gold wafer and is a measure for the highest conductivity that can be measured using this approach. **c** NCM amplitude map for a different area of the same polymer-stabilized ACC sample showing a large structure of several μm in diameter and **d** corresponding C-AFM results, showing good conductivity across a measurement distance of several 100 nm (more details are shown in Supplementary Fig. 24). **e** NCM amplitude map for a polymer-free ACC sample and **f** corresponding C-AFM results, showing good conductivity up to a Z-height of 100 nm (more details are shown in Supplementary Fig. 23).

ions, mostly bicarbonate at the investigated pH levels, are present on the surface of the dense liquid to balance out remaining charges and form counter-ion clouds in the interstices constituted by the mother phase, alongside hydroxide ions. In this liquid phase, the (bi)carbonate and hydroxide populations are equilibrated according to the constant, basic pH level of 9.8. Upon aggregation and coalescence of the nanodroplets, however, these species are kinetically entrapped by internalization of the surfaces within the growing liquid-like precursor. Due to the colloidal stabilization of the nanodroplets by the polycarboxylate, large loose agglomerates can form as a starting point, which have a larger internalized surface than non-stabilized, smaller ones forming in the absence of polymers, i.e., without colloidal stabilization. Upon ongoing dehydration and solidification of the dense liquid with the internalized bicarbonate (by kinetic entrapment), unscreened calcium-bicarbonate interactions destabilize the latter species, which transforms into carbonate and water, together with

entrapped hydroxide ions, giving rise to locally calcium-deficient environments. These are globally charge-balanced by additional calcium ions binding to the polymers, which has given rise to the counter-ion charge balancing and subsequent bicarbonate entrapment, as described above. Since the bicarbonate entrapment in DLP necessitates the generation of localized charges in solid ACC leading to local stoichiometric mismatches, a significant barrier for solid nucleation is imposed. The local deviations from the 1:1 calcium carbonate composition, alongside significant amounts of incorporated polymer, thus contribute to the strong inhibition of the nucleation of solids in the presence of polycarboxylates and, later, their crystallization.

We propose that the interfaces of the DLP-nanodroplets, which are decorated with polycarboxylates and counter-ions, mostly bicarbonate and hydroxide, become internalized in the growing liquid-like mineral via aggregation and coalescence. Upon dehydration and solidification of the DLP, the previous interface remains in solid ACC as a

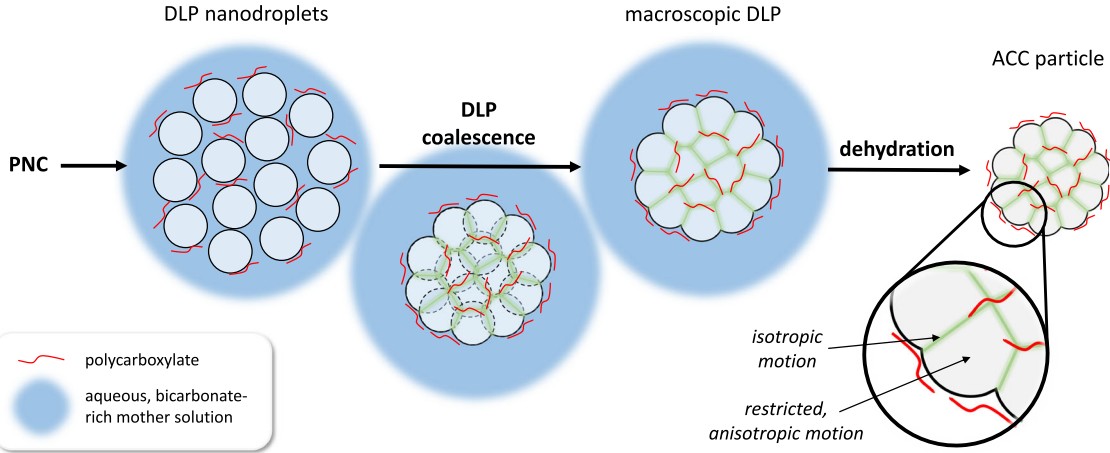

**Fig. 5 | Cartoon depicting the proposed formation mechanism of distinct water environments present in ACC (not to scale).** Solute prenucleation clusters (PNC) undergo phase separation and form nanodroplets of a dense liquid phase (DLP) of the mineral. The polycarboxylate binds to calcium ions on their surface and colloidally stabilizes them. Eventually, the nanodroplets aggregate to reduce their interfacial free energy, and the species present on the surface of the droplets, as well as counter-ions present in the mother solution (mostly bicarbonate, and also hydroxide ions at pH > 9, shown in blue), are entrapped in the growing liquid-like mineral precursor via interface internalization (green areas). Due to the high polymer concentration and the different chemical environments on the surface of the DLP droplets, the distinct chemical environments will remain in the growing DLP and be transferred into solid ACC upon dehydration and solidification. As shown by MAS NMR, one of these environments is rigid and allows only restricted, anisotropic motion of the water molecules. We attribute this environment to the bulk of the original DLP nanodroplets. The other environment remains from the imperfect coalescence of the DLP nanodroplets and hence consists of water molecules undergoing isotropic motion and kinetically entrapped hydroxide ions, which form a network across the mineral precursor.

distinct chemical environment in which water molecules can undergo isotropic motion. This environment forms a network throughout the bulk of ACC and enables the observed conductivity. The other, rigid environment allows only anisotropic motion of water molecules and arises from the bulk of the phase-separated nanodroplets. The structural water in this environment comes from the coordination waters of calcium and carbonate ions that remain in the bulk mineral. Note that the two environments do not form separate phases, i.e., there are no phase boundaries, at least not on a scale larger than several nm. This interpretation agrees remarkably well with previous investigations of ACC formation mechanisms[23] and offers a unifying explanation for the results from MAS NMR, C-AFM, TGA, and potentiometric titrations.

In summary, polycarboxylate additives facilitate the kinetic entrapment of bicarbonate ions in liquid-mineral precursors during the aggregation of colloidally stabilized DLP nanodroplets. This effect seems to play a crucial role in polycarboxylate-controlled mineralization. Our findings imply that the notions of the PNC pathway[23], previously established for additive-free systems, can be extended for polycarboxylate-controlled scenarios. The polymer-facilitated entrapment of bicarbonate ions in the DLP precursor effectively leads to local stoichiometric mismatches in solid ACC, which explains the strong inhibition of solid nucleation and crystallization by polycarboxylates. While further research is necessary to elucidate the process of (locally) restoring stoichiometry upon crystallization, it is clear that the effect of polycarboxylates is leveraged by the entrapment of bicarbonate ions in the DLP precursor. In biomineralization, this effect can be of even larger importance, as it takes place in seawaters at lower pH values than explored herein and, thereby, at higher bicarbonate concentrations.

MAS NMR experiments reveal two distinct water environments in ACC, the presence of which can be explained by the PNC pathway as well. Interfaces of DLP nanodroplets, colloidally stabilized by polycarboxylates, are carried over to ACC upon aggregation, coalescence, and dehydration of the DLP precursor, accompanied by the entrapment of ions from the mother solution. This creates an environment in which water molecules undergo isotropic motion and form a mobile network throughout the bulk of ACC. Dissolved hydroxide ions in this network are the charge carriers responsible for electrical conductivity,

which was detected by C-AFM across several 100 nm. ACC, therefore, is a mineral ion conductor that can potentially be used in electrochemical devices.

## Methods
### Materials

All solutions were prepared using Milli-Q water that was degassed by bubbling $N_2$ through the solution overnight to remove dissolved $CO_2$. 20 mM $CaCl_2$ and 20 mM NaOH solutions were prepared by dilution of $CaCl_2$ stock solution (1.0 M, VWR, AVS Titrinorm volumetric solution) and NaOH stock solution (0.1 N, Carl Roth), respectively. Carbonate buffer solutions were freshly prepared before each experiment by dissolving $Na_2CO_3$ (Sigma-Aldrich, ACS grade, 99.95–100.05%) and $NaHCO_3$ (Sigma-Aldrich, ACS grade, >99.7%) in appropriate ratios for the desired pH value (pH 9.0, 9.4, 9.8, and 10.2). For $^{13}C$ enriched samples, $Na_2^{13}CO_3$ (99% $^{13}C$, Cambridge Isotope Laboratories) and $NaH^{13}CO_3$ (99% $^{13}C$, Cambridge Isotope Laboratories) were used. Prior to titration experiments, the pH values of the buffer solutions were adjusted by using small quantities of 0.1 N NaOH or 0.1 N HCl (Carl Roth), if necessary. For ionic strength-corrected solutions, NaCl (Carl Roth, ≥99,5%) was used. Polymer stock solutions (100 mg/L) were prepared by dissolution of poly(L-aspartic acid sodium salt) (PAsp, Alamanda Polymers, MW = 6800 Da, PD = 1.04), poly(L-glutamic acid sodium salt) (PGlu, Alamanda Polymers, MW = 7500 Da, PD = 1.03), or poly(acrylic acid), 2-(Dodecylthiocarbonothioylthio)-2-methylpropanoic acid (DDMAT)-terminated, (PAA, Sigma Aldrich, MW = 8000 Da, PD < 1.1) in $CO_2$-free Milli-Q water. Detailed characterization of the used polymers is presented in an earlier study[43]. The polymer stock solutions were stored at 5 °C and were used for no longer than 2 days. Prior to the preparation of PAsp stock solutions, the polymer was purified twice by diafiltration using a centrifugal filter device with a MWCO of 3 kDa (Merck Amicon® Ultra-15). This removes small molecular weight fractions from the polymer solutions that can penetrate the membrane of the calcium ion selective electrode (Ca-ISE) and therefore affect the recorded potential or damage the electrode[43]. Polymer-containing carbonate solutions were freshly prepared before each experiment using polymer stock solution, $Na_2CO_3$, and $NaHCO_3$ in appropriate ratios.

## Titration setup

A commercial automated titration setup (Metrohm Titrando or Metrohm OMNIS) controlled by computer software (Metrohm tiamo or Metrohm OMNIS) was used for the titration experiments. A titration device (905 Titrando or OMNIS Titrator) controls two dosing devices (800 Dosino or OMNIS Titration / Dosing Modules) for the precise addition of $CaCl_2$ and NaOH solutions. The calcium potential was monitored using a calcium ion selective electrode (Ca-ISE; Metrohm, No. 6.0508.110) and the pH was measured using a pH electrode (Metrohm, No. 6.0256.100). In addition, the transmission of the solution was measured with an optrode (Metrohm, No. 6.1115.000) using a wavelength of 660 nm. The inner reference system of the pH electrode was used as a reference electrode for the calcium electrode.

## Titration experiments

**Characterization of additive effects.** Calibration of the pH electrode was carried out at least twice per week using pH buffers from Mettler Toledo with pH 4.01 (No. 51302069), 7.00 (No. 51302047), and 9.21 (No. 51302070). The Ca-ISE was calibrated at least once per day by dosing 20 mM $CaCl_2$ solution at a rate of 0.01 mL/min into 50 mL Milli-Q water at the same pH value as the actual experiment. For experiments with polymers, the Ca-ISE was calibrated before and after each measurement to confirm the reproducibility of the electrode signal after exposition to the polymers and to exclude the possibility of damage to the electrode[43]. The calibration was carried out for 90 min. Both pH and calcium ion selective electrodes were regenerated twice a week by stirring for 2 h in 100 mM HCl (pH electrode) and 100 mM $CaCl_2$ (Ca-ISE).

Titration experiments were carried out by adding 20 mM $CaCl_2$ solution at a rate of 0.02 mL/min (pH 9.0) or 0.01 mL/min (pH 9.4, 9.8, and 10.2) into 50 mL of 10 mM carbonate buffer at the desired pH value. All experiments were performed in a sealed beaker under an $N_2$ shower, saturated with water vapor, to prevent diffusion of $CO_2$ into the solution. During the experiment, the pH was kept constant by automatic addition using 20 mM NaOH. Polymer-containing experiments were performed in the same way using 50 mL of 10 mM carbonate solution containing 10 mg/L of a polymer. For each pH value and each polymer, at least 3 independent experiments were performed to ensure reproducibility.

After each titration experiment, the beaker and electrodes were washed two times using hydrochloric acid (3%, prepared by dissolution of 32% HCl, Carl Roth Rotipuran® p.a.) to remove traces of mineral precipitate of the vessel, electrodes, and dosing tips. After washing with HCl, the equipment was rinsed using Milli-Q water and dried with dust-free tissue paper.

**Determination of bicarbonate binding.** The electrode calibrations were performed in ionic strength-adjusted solutions by dosing 20 mM $CaCl_2$ into 20 mL of 14.5 mM NaCl solution (see Section 2.1.1 in the SI). Thus, actual ion products were determined from all experiments[69]. Titration experiments were performed using 0.01 g/L (PAA) or 0.1 g/L (PAsp and PGlu) polymer concentrations at pH 9.8. The experiments were carried out by dosing 20 mM $CaCl_2$ with 0.01 mL/min into 20 mL of polymer-containing carbonate buffer solution (10 mM carbonate concentration). The pH was held constant by the addition of 10 mM NaOH and 10 mM HCl, respectively. The reaction vessel was flushed with $N_2$ for 10 min prior to the start of the experiment to remove $CO_2$ that could potentially diffuse into the solution. During the experiment, no $N_2$ shower was applied.

**Isolation of pre- and postnucleation calcium carbonate samples**
ACC samples were directly isolated from titration experiments[44]. Titration experiments were performed at pH 9.4 (see the previous subsection). At the desired time, the sample was isolated by pouring the reaction solution (roughly 50 mL) into 1.5 L absolute ethanol (99.95%, VWR No. 20820.293) under vigorous stirring. After 30 min of stirring, the beaker was covered with parafilm and left standing for 90 min to let the particles sediment. Then, most of the ethanol was removed by decantation and the particles were isolated from the remaining solution (usually a few 100 mL) by centrifugation for 10 min at 7000 $g$. The particles were washed with 50 mL of ethanol, followed by another centrifugation step. The procedure was repeated once more with ethanol and once with acetone (p.a. >99.98%, VWR No. 20066.296). Then, the particles were dried for 2 h at 40 °C in a vacuum. The samples were stored in a vacuum until characterization was performed. For samples isolated after 16 h, no quenching was performed, and the particles were directly isolated by centrifugation. The washing and drying steps were performed as described for ethanol-quenched samples.

## Preparation of MAS NMR and C-AFM samples

**Pure ACC (Abbreviation: ACC).** 2 L ethanol (99.9%, VWR No. 20820.293) was stirred in a plastic beaker. 50 mL of 50 mM $CaCl_2$ solution was added and after stirring for 2 min, 50 mL of 50 mM $Na_2CO_3$ solution was added. The beaker was sealed with parafilm and after stirring for 30 min, the stirrer was removed, and the beaker was resealed. The beaker was left standing for 30 min, then the supernatant was decanted, and the remaining opaque sediment of ACC was isolated by centrifugation at 6000 $g$. The sediment was washed twice with 50 mL pure ethanol (VWR No. 20820.293), followed by 50 mL pure acetone (VWR No. 20066.296), and then stored in pure acetone. ATR-FTIR and TGA analysis of the product is shown in Supplementary Figs. 6 and 8, respectively.

**PAsp stabilized proto-vaterite ACC (Abbreviation: PAsp_ACC).** 100 mM $CaCl_2$ solution was added at a rate of 0.01 mL/min into 150 mL of 10 mM carbonate buffer containing 0.1 g/L PAsp at pH 9.8. During the experiment, the pH was kept constant by the automatic addition of 100 mM NaOH. The experiment was performed in a sealed beaker to prevent in-diffusion of $CO_2$. After 10,000 s, the reaction was stopped by pouring the reaction solution into 2 L of pure ethanol (99.9%, VWR No. 20820.293) that was stirred in a plastic beaker. The beaker was sealed with parafilm, and after stirring for 30 min, the stirrer was removed, and the beaker was resealed. The beaker was left standing for 60 min, then the supernatant was slowly decanted, and the remaining opaque sediment of ACC was isolated by centrifugation at 6000 $g$ for 15 min. The sediment was washed with ethanol (VWR No. 20820.293), followed by another centrifugation for 15 min. The procedure was repeated once more with ethanol and once with pure acetone (VWR No. 20066.296). The ACC was dried and stored at 40 °C in a vacuum until the filling of the rotor. The synthesis yielded 20 mg of ACC per batch. Five experiments were performed to gather enough material to fill a 4 mm MAS NMR rotor. The quality of the ACC was confirmed by ATR-FTIR spectroscopy for each batch. After each titration experiment, the beaker and electrodes were washed two times using acetic acid (10%) to remove traces of mineral precipitate from the vessel, electrodes, and dosing tips. After washing with acetic acid, the equipment was rinsed using Milli-Q water and dried with dust-free tissue paper. ATR-FTIR and TGA analysis of the product is shown in Supplementary Figs. 6 and 8, respectively.

**PAsp calcium salt (Abbreviation: PAsp_Ca).** 76.0 mg of PAsp were dissolved in 40 mL of Milli-Q water. Then, 7.5 mL of 1.0 M $CaCl_2$ standard solution (VWR, No. 190646 K) was slowly added to the stirred polymer solution while the pH was monitored using a pH electrode (Metrohm No. 6.0256.100). The pH dropped from pH 9.5 to pH 8.2 during the addition of the calcium solution. The pH was then set to pH 10.0 by slowly adding 0.1 M NaOH standard solution (Roth, No. K020.1). After stirring for 5 min, the solution was added dropwise into 250 mL of a 1:1 (v:v) solution of absolute EtOH (VWR, ≥99.8%, AnalaR NORMAPUR® ACS) and absolute acetone (VWR, AnalaR NORMAPUR®

ACS), which was stirred in a 1 L glass beaker. After stirring for 10 min, the cloudy white solid was isolated by centrifugation at 7000 $g$ for 15 min. The obtained white sediment was resuspended in approx. 100 mL abs. EtOH and centrifuged again. This step was repeated one more time. Then, the product was dried at 40 °C in a vacuum for 60 min. The synthesis yielded 78.1 mg of product. NMR, ATR-FTIR, and TGA analysis of the product is shown in Supplementary Figs. 2, 6, and 8, respectively.

Additional inductively coupled plasma optical emission spectrometry (ICP-OES) analysis was performed on a Spectro Arcos spectrometer. Samples were dissolved in 4% $HNO_3$ and measurements and calibrations were performed in 4% $HNO_3$. Calibration solutions for 5-point calibration (1 to 5 ppm) were prepared from Ca and Na standard solutions (1000 ppm ICP standard solutions, Carl Roth). For the pure PAsp (purchased as a sodium salt), a value of 0.65 bound $Na^+$ per carboxyl group was determined by ICP-OES analysis. The PAsp_Ca sample showed 0.38 $Ca^{2+}$ bound per carboxy group, while no significant amounts of $Na^+$ were determined, confirming the successful ion exchange.

**Pure PAsp (Abbreviation: PAsp).** PAsp was used as purchased without further purification. If no experiments were performed, the salt was stored in inert gas atmosphere at −20 °C (for long-term storage) or in a vacuum (short-term storage). NMR, ATR-FTIR, and TGA analysis of the product is shown in Supplementary Figs. 2, 6, and 8, respectively.

**PAsp stabilized disordered ACC (Abbreviation: PAsp_disACC).** 500 mL of a solution of PAsp50 (100 mg/L) and 10 mM carbonate was prepared and set to pH 9.8 with the addition of 200 mM NaOH. Then, $CaCl_2$ solution (200 mM) was added with a rate of 0.4 ml/min to the stirred polymer/carbonate solution using a commercial titration setup (Metrohm Titrando). Simultaneously with the addition of $CaCl_2$, the pH was held constant by counter titration of 200 mM NaOH. The titration was performed under vigorous stirring and under a stream of water-saturated nitrogen to prevent in diffusion of $CO_2$. After 1600 s, the reaction was quenched by pouring the reaction solution into 4 L of pure ethanol (99.9%, VWR No. 20820.293) under vigorous stirring. After stirring for 15 min, the solution was left standing for 30 min. Then, the supernatant was decanted, and the sediment of ACC was centrifuged at 6000 $g$. The solid precipitate was washed with 50 mL of pure ethanol then 50 mL of pure acetone. The ACC was stored in pure acetone. NMR, ATR-FTIR, and TGA analysis of the product are shown in Supplementary Figs. 2, 6, and 8, respectively.

In preparation for MAS NMR experiments, PAsp_disACC was isolated from the acetone dispersion by centrifugation at 6000 $g$ and was immediately dried at 40 °C in a vacuum for 1 h. After drying, the ACC powder was directly packed into a 4 mm rotor. Then, the packed (but still open) rotor was again dried for 30 min at 40 °C in a vacuum, followed by immediate closing with a $ZrO_2$ Cap. For shipment to Konstanz, rotors were placed in an Eppendorf tube, which was flushed with $N_2$ and sealed with parafilm and itself placed in a Falcon tube, which was again sealed with parafilm in $N_2$ atmosphere. Rotors were only taken out of their packaging immediately before they were inserted into the MAS NMR probe, which had already been flushed and brought to the appropriate temperature with nitrogen gas.

**Monohydrocalcite (Abbreviation: MHC).** 50 mL of a solution of 0.077 M $Na_2CO_3$ (Sigma-Aldrich, 99.95−100.05%) was slowly poured into 250 mL of a stirred solution of 0.052 M $CaCl_2 \cdot 2H_2O$ (Sigma-Aldrich, >99%) and 0.010 M $MgCl_2 \cdot 6H_2O$ (Sigma-Aldrich, >99%). The solution was stirred at 25 °C for 24 h. The product was isolated by centrifugation and washed two times with Milli-Q water and once with pure ethanol (99.95%, VWR No. 20820.293). The product was dried at 40 °C in a vacuum overnight. 1.9 g of a white solid was obtained[70].

Sample analysis with ATR-IR, XRD, TGA, and DSC is shown in Supplementary Fig. 25.

## Sample characterization

**TGA and TGA-MS-IR.** Thermogravimetric analysis (TGA) was performed on a Netzsch STA 409 PC LUXX in an oxidative atmosphere ($Ar/O_2$ 80:20 v:v) and a heating rate of 5 K/min. Analysis of decomposition gases using mass spectrometry and infrared spectroscopy (TGA-MS-IR) was performed by coupling the TGA device with a Netzsch QMS 403 D Aelos mass spectrometer and a Bruker Invenio S FTIR spectrometer with an external gas cell. Before each measurement, the empty $Al_2O_3$ sample pan was heated to 1000 °C to remove any remaining impurities. Measurements were performed with at least 10 mg of sample and prior to TGA analysis, samples were dried at 40 °C in vacuum overnight.

**Attenuated total reflection Fourier-transform infrared.** Attenuated total reflection Fourier-transform infrared (ATR-FTIR) spectroscopy was performed using a Bruker Tensor 27 or Bruker Vertex 70 v spectrometer. The absorbance of the sample was measured from 4000 to 650 $cm^{-1}$ with a resolution of 1 $cm^{-1}$. The samples were directly placed as a powder on the ATR unit for measurements.

**Scanning electron microscopy.** Scanning electron microscopy (SEM) was performed on a JEOL JSM-6700F SEM. Samples were coated with a 5 nm thick layer of gold (Cressington 108auto) prior to analysis.

**X-ray powder diffraction.** X-ray powder diffraction measurements (XRD) were performed on an STOE Stadi P diffractometer in transmission using Cu-Kα radiation ($\lambda = 1.54060$ Å, Generator: 40 kV, 30 mA) and a curved Ge (111) monochromator. Sample powders were fixed between X-ray amorphous foils for measurement.

**Atomic force microscopy and conductive-AFM.** Atomic force microscopy (AFM) and conductive AFM (C-AFM) were performed using a Park Systems NX 10 microscope in a glovebox in $N_2$ atmosphere. All measurements were performed using a CDT-FMR 10M_T conductive cantilever.

For sample preparation, a Si wafer (1 × 1 cm, Ted Pella) was coated with a 30 nm Au layer using a Cressington 108auto coater. Then, the wafer was coated with the sample dispersion (usually, the ACC samples were obtained as dispersion in acetone after synthesis, see the previous subsection on sample preparation) using a spin coater. Depending on the concentration of particles, the spin-coating process was repeated several times to deposit an appropriate number of particles. The Si wafer was then placed on the C-AFM sample holder disk and fixed with silver conductive paint to ensure good conductivity between the Au layer and the AFM holder (see Supplementary Fig. 36a in Section 4.1 in the Supplementary information). Prior to C-AFM analysis, the samples were stored in the glovebox and locked in a vacuum overnight to remove surface adsorbed water, as surface adsorbed water must be removed to the extent that no conductivity is caused by a water layer on the surface of the particles. Drying the samples in the glovebox lock proved to be sufficient for this, as shown by vaterite reference samples.

In AFM experiments, non-contact mode (NCM) was used to find suitable spots for C-AFM analysis. Then, C-AFM measurements were performed in spectroscopy mode using a sample bias from −1 to 1 V, while the cantilever was lifted between the measurement points. A set point value (pressing force for measurement) of 3 V and settling time of 500 ms was used. The detailed steps for the evaluation of experimental data are described in Section 4.2 in the Supplementary information.

**MAS NMR**. Magic-angle spinning nuclear magnetic resonance (MAS NMR) experiments were performed on a Bruker Avance III 400 MHz spectrometer, which was equipped with a wide-bore magnet and 4-mm MAS HX probe. Halfway through the project, the original probe, which was more than 10 years old, was replaced by a new probe. This improved the sensitivity of the $^{13}$C channel by a factor of 2.5. The sample temperature was controlled using a BCU-Xtreme cooling unit. The temperature of the incoming nitrogen gas and the flow rate were adjusted to the spinning frequency in accordance with a temperature calibration performed with KBr[71]. Chemical shifts were referenced to TMS using adamantane as an indirect reference; the field was adjusted such that the $^{13}$C low-field signal was observed at 38.48 ppm[72].

The directly detected $^1$H spectra were acquired with a 90°-pulse of 3 μs. Acquisition times were 33 ms for the sample of PAsp and 121 ms for the samples of PAsp-stabilized ACC. The dwell time was 4 μs. Typically, 32 scans were averaged with recycle delays between 2.5 and 6.5 s. Before Fourier transformation, transients were zero-filled up to 65536 points. No window function was applied.

The $^{13}$C direct excitation spectra were acquired with a 90°-pulse of 4 μs. The dwell time was 12.133 μs, and the acquisition time was 50 ms. 1648 scans were averaged with a recycle delay of 280 s. During acquisition, small-phase incremental alternation with 64 steps (SPINAL-64) heteronuclear decoupling[73] was applied on the $^1$H channel at a nutation frequency of 83 kHz. A $^{13}$C background signal from the probe did not interfere with the spectral features from ACC and PAsp and was removed by baseline subtraction. Transients were zero-filled up to 16384 points. No window function was applied.

$^1$H–$^{13}$C cross-polarization (CP)[74] experiments were done with a 90° $^1$H pulse of 3 μs. On the old probe, during the contract period, an RF power of 75 W (48 kHz) was applied on the $^{13}$C channel and a 70–100% ramp up to 85 W (70 kHz) was applied on the $^1$H channel. On the new probe, an RF power of 55 W (50 kHz) was applied on the $^{13}$C channel and a 70–100% ramp up to 85 W (73 kHz) was applied on $^1$H channel. The contact time was optimized for the carbonate signal of ACC at 4 ms. During acquisition, SPINAL-64 decoupling was applied at a nutation frequency of 83 kHz. The CP spectra of PAsp and PAsp_Ca are averages of 16384 scans, with a dwell time of 16 μs and acquisition time of 33 ms. CP spectra of PAsp-stabilized ACC were averages of 1024 scans, with a dwell time of 12.133 μs and an acquisition time of 50 ms. To enable quantitative comparison the recycle delays were set to $1.26 \cdot T_1$, which amounted to 4 and 6.5 s at 25 and −25 °C, respectively. Transients were zero-filled up to 16384 points. No window function was applied.

$^1$H–$^{13}$C correlation spectra, without homonuclear decoupling during the evolution period, were recorded with the wideline-separation (WISE) sequence[52], also referred to as heteronuclear correlation (HETCOR). The sequence starts with a 90° $^1$H pulse of 3 μs. The States-TPPI acquisition scheme was used[75,76]. The evolution time was incremented in 128 steps of 4 μs, which gives a resolution of 5 ppm in the $^1$H dimension. During the contact period, an RF power of 70 W was applied on the $^{13}$C channel and a 70–100 % ramp up to 60 W was applied on the $^1$H channel. The contact time was 1 ms. During acquisition, SPINAL-64 decoupling was applied at a nutation frequency of 83 kHz. The WISE spectrum at 5 kHz spinning frequency was an average of 512 scans, and the WISE spectrum at 10 kHz was an average of 256 scans. The recycle delay was 2.5 s. Zero-filling was applied up to 8192 points in the direct dimension and up to 2048 points in the indirect dimension. In the direct dimension, an exponential window function with a width of 80 Hz was applied to the 5-kHz WISE spectrum and an exponential window function with a width of 65 Hz was applied to the 10-kHz WISE spectrum. In the indirect dimension, no window function was applied.

To record $^1$H–$^{13}$C HETCOR spectra with homonuclear decoupling, the frequency-switched Lee-Goldburg (FSLG) sequence[77–79] was applied during the evolution period. A 90° $^1$H pulse of 3 μs was used. The evolution time was incremented in 64 steps of 78.4 μs, which gives

a resolution of 1 ppm in the $^1$H dimension. During the contact period, an RF power of 75 W (48 kHz) was applied on the $^{13}$C channel, and a 70–100% ramp up to 85 W (70 kHz) was applied on the $^1$H channel (old probe). The contact time was 400 μs. During acquisition, SPINAL-64 decoupling was applied at a nutation frequency of 83 kHz. The 128 averages were done with a recycle delay of 3 s at a 10 kHz spinning frequency. Zero-filling was applied up to 8192 points in the direct dimension and up to 1024 points in the indirect dimension. In the direct dimension, an exponential window function with a width of 80 Hz was applied. In the indirect dimension, an exponential window function with a width of 100 Hz was applied. In experiments, the scaling of the evolution of the chemical shift under FSLG decoupling often deviates from the theoretical value of $1/\sqrt{3}$. To determine the scaling, each measurement session included a series of FSLG HETCOR spectra, recorded with different carrier frequencies, of a reference sample of microcrystalline α-Gly. Experimental scaling factors were typically found between 0.515 and 0.527. In addition, an offset of 2700–3000 Hz had to be included to align the observed $^1$H chemical shifts with the known $^1$H chemical shifts of α-Gly (8.0 ppm for the NH moiety, 3.8 and 2.6 ppm for the two $^1$Hs of $C_\alpha$, following Stievano et al.[80]).

## Synthesis of reference samples for C-AFM measurements

**Synthesis of gold nanoparticles.** Au nanoparticles were used as conductive reference material for C-AFM investigations and synthesized according to a standard protocol by Huang et al.[81] 181 mg Cetyltrimethylammonium bromide (CTAB, Fluka Analytical, >99.0 %) were dissolved in 32 ml MilliQ water in a 50 ml SCHOTT vial. Then, 165 μl of a 0.1 M trisodium citrate ($Na_3C_6H_5O_7$, anhydrous, Merck, EMPROVE® EXPERT) solution and 825 μl of 10 mM hydrogen tetra-chloroaurate trihydrate ($HAuCl_4 \cdot 3H_2O$, Sigma-Aldrich, 99.9 %) solution were added. The vial was wrapped with aluminum foil and loaded in an oven at 110 °C. After 6 h the sample was taken from the oven to cool down. The resulting nanoparticles were isolated by centrifugation at 7000 $g$ for 20 min. The particles were then washed two times with 1 M CTAB solution and redispersed in MilliQ water. Then, the particles were washed several times with water to remove excess CTAB from the particles.

**Synthesis of vaterite nanoparticles.** 2.5 mmol of calcium chloride dihydrate (Sigma-Aldrich, ACS grade, >99%) were dissolved in 25 mL of ethylene glycol (>99%, Carl Roth) by ultrasonication (Elmasonic P, 80 kHz, 80% power, 40 °C, sweep mode) for 30 min. Then, a dispersion of 5 mmol sodium bicarbonate (Sigma-Aldrich, ACS grade, >99.7%) in ethylene glycol was added under stirring and the resulting dispersion was treated for 30 min by ultrasonication (80 kHz, 90% power, 40 °C, sweep mode)[82]. The particles were isolated by centrifugation for 30 min at 7000 $g$ and washed twice with Milli-Q water and once with ethanol (99.95%, VWR No. 20820.293). The product was stored in ethanol. ATR-FTIR characterization of the dried particles conformed to pure vaterite.

## Data availability

All data generated in this study have been deposited in the Zenodo repository. Data are also available from the corresponding authors upon request.

## Code availability

Scripts used for numerical simulations have been made available in the Spinach library.

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

## Acknowledgements

This research was supported by the Deutsche Forschungsgemeinschaft through the Emmy Noether Program awarded to GM (project number 321027114). The authors thank Sarah Busse for providing gold nano-particles, Thomas Herzog for help with AFM measurements, Stella Kittel for help with ICP-OES measurements, Katharina Nolte for support during TGA-MS-IR experiments, and David McDonogh for help with XRD measurements, and Ilya Kuprov and Albert A. Smith-Penzel for advice and stimulating discussions.

## Author contributions

M.B.G. performed the main titration experiments, sample preparations, TGA-MS-IR analysis, and C-AFM and wrote the original paper draft. J.R. performed supplementary titration experiments and SEM and FTIR measurements of isolated samples. D.B. and E.G. performed additional titration experiments. S.V.-K. performed and analyzed the NMR experiments, with support from V.S.R.R. G.M. supervised the NMR spectroscopy, performed numerical simulations, and developed the project idea. D.G. supervised all other experiments and developed the project idea. The paper was written through the contributions of all authors.

## Funding

## Competing interests

There are no competing interests to declare.
