## [Peer Review File · Nature Communications]

Reviewers' Comments:

Reviewer #1:

Remarks to the Author:

Author

Crystalline CaCO₃ While polymer-free ACCs mainly transform into crystalline CaCO₃ via dissolution-reprecipitation pathways, the polymer stabilized DLP/ACC can also transform via pseudomorphic transformation, thereby forming structures with altered shapes and non-equilibrium morphologies. As mentioned earlier, the stoichiometric "mismatch" between the precursor and crystalline CaCO₃ is one reason for the increased kinetic stability of polymer-stabilized mineral precursors.

Referee

Surely, your Reference 17 is interesting; nevertheless, when dealing with the equilibrium shape (ES) of a crystal, the fundamental paper by M. Bienfait and R. Kern cannot be forgotten. In any case, attaining the ES is a complicated path and, personally I think that its thermodynamic is more complex than your "...polymer-free ACCs mainly transform into crystalline CaCO₃ via dissolution-reprecipitation pathways".

M. Bienfait, R. Kern. Bull. Soc. Franç. Minéral. Crist., 87 (1964), p. 604.

Reviewer #2:

Remarks to the Author:

This paper deals with the crystallization of CaCO₃ stabilized by PAsp or PGLu using various techniques like potentiometric titration, C-AFM, TGA-MS, NMR and SEM and relate that to the crystallization inhibition by polycarboxylates. The manuscript shows the incorporation of relatively large amounts of HCO₃⁻ ions, an aspect that indeed previously seems to have been overlooked.

Line 110: the authors state the formal removal of the ions. I understand what they mean but I find the description awkward as it means something like not available for the nucleation process. In line 126 they also use the word removal, again meaning something similar.

Figure 2a shows for the TGA measurements also a significant decrease in weight below 100 deg. C. Do the authors ascribe that to evaporating ethanol or is it still adsorbed water? There is also hardly any signal for the DSC in this range while for higher temperature the signals correspond. Is the binding energy thus really that low? Fig. S16 suggests that it is water but the authors also indicate pockets of ethanol in fig. 2. Some clarification would be appreciated.

Around line 200 the authors indicate that there is also carbon 12 CO₂ released and ascribe that to the polymer. Might it also be from the remaining Na₂CO₃ in carbon 13 Na₂CO₃ or NaHCO₃?

Around line 219 the authors indicate that no Na salts are incorporated and provide XRD data to support that. However, the XRD spectrum in fig. S17 shows three rather clear but unexplained peaks. Could these be from some unusual complex of Na with one the other components? Or do the authors have another, not mentioned thought about this?

Around line 280 the authors discuss the ACC particles. In the SI they indicate that, although the lateral size is 20-50 nm, the height is "significantly less than 20 nm". From fig. S20 I estimate about 5 nm. This implies a large shape anisotropy of 4 to 10, dependent on the lateral size taken, although the particles are amorphous. This appears rather strange to me. Does it imply a "drying" artefacts, and, if yes, what other consequences that might have had?

They also indicate the effect of "grain boundaries". Although I understand what they mean, the term is inappropriate as a grain boundary is a region between differently oriented crystalline

regions in a polycrystalline material. Typically, grain boundaries in solids do show an enhanced conductivity and not a lower one, like here. I suggest that the authors describe that they mean the loose contact between the various particles in the deposited sample.

Around line 290 the authors state "All of these structures exhibit electrical conductivity across length scales that are orders of magnitude larger than single ACC particles". I found this rather confusing. If I am right, they mean that the height reaching similar conductivity as for the ACC is much larger. This also begs the question how it is possible to have conductivity over such a large distance. This cannot be tunneling, I suppose. Indeed the tip induces a large field strength but is that sufficient?

In line 321 the authors indicate "While our study does not provide a direct explanation for a mechanism how low amounts of polycarboxylates facilitate considerable bicarbonate binding, ...". I would say this is the crux of the matter.

Finally, as stated in the introductory part and repeated in other words in the final lines of the manuscript, in my view the statement that potentially designed new materials based on proton conductive amorphous minerals will be furthered is rather far-fetched.

Overall, the SI is difficult to read by referring hence and forth, while if one consults the SI, one expects to be informed about relevant details at one spot. One typo is on page 4, second paragraph: "... sealed beaked ..." where beaker is meant, I suppose.

On page 7 of the SI the authors indicate "samples were dried at 40 °C in vacuum overnight to remove surface adsorbed water". Something similar is said on page 7 for storing overnight in the glovebox lock. I understand the low temperature (40 or room temperature) but I think, from experience, this is insufficient to remove adsorbed water.

This brings me to an overall assessment from my side. In my opinion the experiments and interpretation have been done carefully, apart from the aspects mentioned. These aspects need attention. As there is no evident suggestion for a mechanism of inhibition, I leave the question whether the results are sufficient to warrant publication in Nature Communications to the editor.

Reviewer #3:

Remarks to the Author:

The authors quantitatively analyzed the incorporation of bicarbonate into ACC in the presence/absence of polymer additives. I read this manuscript with great interest because I am also curious about why a small fraction of polyelectrolyte can effectively stabilize a relatively large amount of ACC. I was impressed by the authors' designs in the experiments, which provided solid evidence to support their conclusions. To further improve the quality of this work, the following issues should be considered.

1. The conductivity is an important property of ACC as a material, and thus should be discussed more prudently. No evidence has been provided to reveal the phase of the large grain in Fig. 3c (SEM-Raman can be helpful). I also doubt if proton is the only species that is responsible for the conductivity; the current C-AFM tests remain semi-quantitative, and possibly further characterizations based on ACC samples with larger size and regular shapes could help.
2. There is essentially no direct comparison in the C-AFM tests between the ACC with/without bicarbonate incorporation. It can be seen from Fig. 2a that the temperature at which bicarbonate decomposes is lower than that for the crystallization of ACC. This allows the elimination of all/most of the bicarbonates in the sample while maintains the amorphous phase. Then it is possible to directly compare the conductivity of two samples: ACC with/without bicarbonate.
3. In Fig. 2d, the small foot is assigned to bicarbonate. Yet the authors commented "However, its relative intensity is a few percent and it is therefore unlikely to be the bicarbonate species detected in the dense liquid phase and in the precipitate by TGA". A contradiction is proposed but without proper discussion. It is also possible to obtain a hint by comparing the NMR of ACC with/without the incorporation of bicarbonate.
4. "In polymer-free experiments, carbonate ions and calcium ions are bound in PNCs, resulting in

the formal removal of carbonate ions from the buffer equilibrium ($\text{HCO}_3^- \rightleftharpoons \text{H}^+ + \text{CO}_3^{2-}$) and release of H^+ ions that are then neutralized by automatic NaOH addition... Thus, from the NaOH addition, the removal of carbonate species from the buffer equilibrium upon addition of calcium due to ion association can be calculated." I found these sentences misleading that I was thinking the authors calculated the amount of carbonate ions simply based on this equation: $n(\text{NaOH}) = n(\text{H}^+ \text{ ion release}) = n(\text{carbonate bound by Ca})$. This equation is obviously not correct because the equilibrium shifts as the total amount of carbonate species (carbonate and bicarbonate) decreases. It seems the authors have already taken this into account (SI, 2.3.3 and 2.3.4). Anyway, I suggest that these sentences should be rephrased.

5. Fig. 1b has not been properly cited.

6. Inconsistency of the abbreviations throughout the manuscript.

7. The scale factor maximized at $\text{pH} = 9.0$. What is the rationale that the "specific pH value" is set to 9.8?

8. The vaterite nanoparticles, which were not stable in water, were washed with deionized water. The authors should provide the evidence to show its phase (instead of "data not shown").

9. The SI file is terribly formatted.

Point-by-point Replies (NCOMMS-23-05524)

Herein, our replies appear indented and italicized, corresponding changes are highlighted in yellow in separate files of manuscript and SI.

Reviewer #1:

Author: Crystalline CaCO₃ While polymer-free ACCs mainly transform into crystalline CaCO₃ via dissolution-reprecipitation pathways, the polymer stabilized DLP/ACC can also transform via pseudomorphic transformation, thereby forming structures with altered shapes and non-equilibrium morphologies. As mentioned earlier, the stoichiometric “mismatch” between the precursor and crystalline CaCO₃ is one reason for the increased kinetic stability of polymer-stabilized mineral precursors.

Referee: Surely, your Reference 17 is interesting; nevertheless, when dealing with the equilibrium shape (ES) of a crystal, the fundamental paper by M. Bienfait and R. Kern cannot be forgotten. In any case, attaining the ES is a complicated path and, personally I think that its thermodynamic is more complex than your “...polymer-free ACCs mainly transform into crystalline CaCO₃ via dissolution-reprecipitation pathways”. M. Bienfait, R. Kern. Bull. Soc. Franç. Minéral. Crist., 87 (1964), p. 604.

Author Reply: We agree that the paper is certainly of relevance for describing the equilibrium shapes of crystals, especially when dealing with transformation of polymer-free ACCs. However, our manuscript mainly deals with the formation of polymer-containing amorphous mineral precursors, rather than polymer-free crystalline materials. As described in our manuscript (see also Gower et. al., Journal of Structural Biology: X 6, 100059, 2022), the minerals formed from these precursors often form nonequilibrium morphologies.

We believe that the aspects discussed in the suggested paper by Bienfait et.al. are not essential to our study to an extent that warrants a citation. Since the cited paper is in French and we perhaps missed an important point, however, we would be most happy to include the citation in our manuscript if the reviewer can provide more details on the importance of the paper for our study.

Reviewer #2:

This paper deals with the crystallization of CaCO₃ stabilized by PAsp or PGlu using various techniques like potentiometric titration, C-AFM, TGA-MS, NMR and SEM and relate that to the crystallization inhibition by polycarboxylates. The manuscript shows the incorporation of relatively large amounts of HCO₃ minus ions, an aspect that indeed previously seems to have been overlooked.

Line 110: the authors state the formal removal of the ions. I understand what they mean but I find the description awkward as it means something like not available for the nucleation process. In line 126 they also use the word removal, again meaning something similar.

Author Reply: We agree the term removal is not optimal when dealing with the processes related to nucleation mechanisms. We have therefore replaced the term and called it “binding”, when talking about the nucleation process.

In the other two instances, we however write “removal of carbonate species from the buffer equilibrium” and we primarily describe the changes in the chemical equilibrium. In this case, in

our view the term removal fits nicely, especially when talking about the LeChatelier principle (e.g., “removal of products from the equilibrium”).

Revisions: Page 4, line 19: Replaced “removal” by “binding”

In fact, ~~it can be shown that~~ at the specific pH value considered here (pH 9.8), the ~~removal~~ binding of 2.33 bicarbonate ions masks the binding of 1 carbonate ion in presence of the polymers, allowing the quantification of the amount of bicarbonate binding from the recorded NaOH addition.

Figure 2a shows for the TGA measurements also a significant decrease in weight below 100 deg. C. Do the authors ascribe that to evaporating ethanol or is it still adsorbed water? There is also hardly any signal for the DSC in this range while for higher temperature the signals correspond. Is the binding energy thus really that low? Fig. S16 suggests that it is water but the authors also indicate pockets of ethanol in fig. 2. Some clarification would be appreciated.

Author Reply: For TGA measurements, samples are dried at 40 °C in vacuum for several hours, so no ethanol is expected to be present in the samples. The significant decrease below 100 °C can be attributed to loosely bound water molecules on the surface of the ACC particles. The water loss in TGA measurements in ACC is, for instance, described in detail in Ihli et.al., Nat. Commun. 2014, 5, 3169. We have added a citation to the paper to further clarify the TGA of ACC.

For NMR, the samples are dried as short as possible due to their instability during measuring, so sometimes there were small amounts of ethanol or acetone visible in the spectra. This was however not detected in most of the samples, while the water loss of ACC in TGA was highly reproducible.

Revisions: Supplementary Information, Page 17, line 5:

a) Pure ACC shows initial water loss (<200 °C), [23] followed by exothermic ACC crystallization [...]

References added:

[23] Ihli, J. et al. Dehydration and crystallization of amorphous calcium carbonate in solution and in air. Nat. Commun. 5, 3169 (2014).

Around line 200 the authors indicate that there is also carbon 12 CO₂ released and ascribe that to the polymer. Might it also be from the remaining Na₂CO₃ in carbon 13 Na₂CO₃ or NaHCO₃?

Author Reply: The ¹³C enriched Na₂CO₃ and NaHCO₃ salts used in our studies had an enrichment of 99% ¹³C. Therefore, only minute amounts of ¹²CO₂ can originate from the salts. As significant amounts of polymer are present in the ACC (roughly 5-10%, dependent on synthesis procedure), we attribute ¹²CO₂ release mostly to the decomposition of polycarboxylate, which shows a natural abundance (99 %) content of ¹²C.

Around line 219 the authors indicate that no Na salts are incorporated and provide XRD data to support that. However, the XRD spectrum in fig. S17 shows three rather clear but unexplained peaks. Could these be from some unusual complex of Na with one the other components? Or do the authors have another, not mentioned thought about this?

Author Reply: We have thoroughly checked the XRD database available to us (Cambridge Structural Database) for the unexplained peaks, but we were unable to assign the reflections. We have also compared the reflections to those of all listed sodium containing compounds (especially oxides that might be formed in TGA) but could not identify any compound. As of now, we do not have an explanation for these reflections.

Around line 280 the authors discuss the ACC particles. In the SI they indicate that, although the lateral size is 20-50 nm, the height is “significantly less than 20 nm”. From fig. S20 I estimate about 5 nm. This implies a large shape anisotropy of 4 to 10, dependent on the lateral size taken, although the particles are amorphous. This appears rather strange to me. Does it imply a “drying” artefacts, and, if yes, what other consequences that might have had?

Author Reply: The observation of shape anisotropy is due to the gel-like properties of the ACC particles. Due to their water content, ACC particles are soft and will “spread” on the wafer after sedimentation. Therefore, an oblate-like structure with significant shape-anisotropy is detected. In addition, the samples were prepared using spin-coating, which exerts corresponding forces on the particles. Similar effects of shape-anisotropy of sedimented amorphous calcium carbonate particles were described in the following studies:

M. B. Gindele, L. V. Steingrube, D. Gebauer, *CrystEngComm* **2021**, *23*, 7938-7943.
S. L. Wolf, L. Caballero, F. Melo, H. Cölfen, *Langmuir* **2016**, *33*, 158-163.

They also indicate the effect of “grain boundaries”. Although I understand what they mean, the term is inappropriate as a grain boundary is a region between differently oriented crystalline regions in a polycrystalline material. Typically, grain boundaries in solids do show an enhanced conductivity and not a lower one, like here. I suggest that the authors describe that they mean the loose contact between the various particles in the deposited sample.

Author Reply: We thank the reviewer for pointing this out. We indeed mean particles in loose contact with each other and have changed the description in our manuscript.

Revisions: Page 12, line 26:

[...], as visible by the AFM height images (Fig. S12), and conductivity is likely poor across ~~grain boundaries~~ particles that are in loose contact with each other.

Around line 290 the authors state “All of these structures exhibit electrical conductivity across length scales that are orders of magnitude larger than single ACC particles”. I found this rather confusing. If I am right, they mean that the height reaching similar conductivity as for the ACC is much larger. This also begs the question how it is possible to have conductivity over such a large distance. This cannot be tunneling, I suppose. Indeed the tip induces a large field strength but is that sufficient?

Author Reply: We agree that this was not clear in the previous version of the manuscript. We have performed additional experiments and are confident that we now have a proper explanation for this effect:

As described in the manuscript, we detect two distinct water environments in the ACC; one of them allows for isotropic motion of water molecules. In addition, we detect hydroxide ions and propose that these constitute the charge carriers in this environment. According to the formation mechanism of the ACC via coalescence of dense liquid nanodroplets, we propose that the mobile water environment is a remainder from the previous interfaces of nanodroplets. These are internalized upon growth via coalescence, additionally resulting in the kinetic

entrapment of hydroxide and bicarbonate ions and in the formation of a network of a dynamic water environment throughout the bulk of ACC particles. This proposed model is consistent with previous studies of calcium carbonate formation, our titration studies, NMR investigations, TGA and the detected conductivity.

In line 321 the authors indicate "While our study does not provide a direct explanation for a mechanism how low amounts of polycarboxylates facilitate considerable bicarbonate binding, ...". I would say this is the crux of the matter.

Author Reply: We would like to emphasize that we have to distinguish between explaining the "mechanism of nucleation inhibition by polycarboxylate", which is provided in our manuscript and the explanation of the "mechanism of bicarbonate binding in presence of polycarboxylate", which we have also rationalized. We copy the corresponding interpretations here, for the detailed results and discussions, please see the revised manuscript:

Revisions: Page 15: Added an additional section discussing the mechanism of ACC formation and inhibition mechanism by bicarbonates, including an additional Figure:

Interpretation of the two environments in ACC

The detection of two distinct water environments is in line with the proposed formation mechanisms of the ACC particles¹¹ (see Fig. 5 for an illustration). The key step of this process is the coalescence of phase separated DLP nanodroplets, which is driven by the reduction of interfacial free energy. The liquid nanodroplets are colloidally stabilized by polycarboxylate molecules binding to calcium ions on their surface, leading to the formation of large, loose agglomerates. These can be visualized by cryogenic electron microscopy and have been erroneously interpreted as solid/liquid interfaces previously.^{11,43} Other ions, mostly bicarbonate at the investigated pH levels, are present on the surface of the dense liquid to balance out remaining charges, and form counter-ion clouds in the interstices constituted by the mother phase, alongside hydroxide ions. In this liquid phase, the (bi)carbonate and hydroxide populations are equilibrated according to the constant, basic pH level of 9.8. Upon aggregation and coalescence of the nanodroplets, however, these species are kinetically entrapped by internalization of the surfaces within the growing liquid-like precursor. Due to the colloidal stabilization of the nanodroplets by the polycarboxylate, large loose agglomerates can form as a starting point, which have a larger internalized surface than non-stabilized, smaller ones forming in absence of polymers, i.e., without colloidal stabilization. Upon ongoing dehydration and solidification of the dense liquid with the internalized bicarbonate (by kinetic entrapment), unscreened calcium-bicarbonate interactions destabilize the latter species, which transforms into carbonate and water, together with entrapped hydroxide ions, giving rise to locally calcium-deficient environments. These are globally charge-balanced by additional calcium ions binding to the polymers, which has given rise to the counter-ion charge balancing and subsequent bicarbonate entrapment, as described above. Since the bicarbonate entrapment in DLP necessitates the generation of localized charges in solid ACC leading to local stoichiometric mismatches, a significant barrier for solid nucleation is imposed. The local deviations from the 1:1 calcium carbonate composition, alongside significant amounts of incorporated polymer, thus contribute to the strong inhibition of the nucleation of solids in presence of polycarboxylates, and later, also their crystallization.

We propose that the interfaces of the DLP-nanodroplets, which are decorated with polycarboxylates and counter-ions, mostly bicarbonate and hydroxide, become internalized in the growing liquid-like mineral via aggregation and coalescence. Upon dehydration and solidification of the DLP, the previous interface remains in solid ACC as a distinct chemical environment, in which water molecules can undergo isotropic motion. This environment forms a network throughout the bulk of ACC, and enables the observed conductivity. The other, rigid environment allows only anisotropic motion of water molecules and arises from the bulk of the phase separated nanodroplets. The structural water in this environment comes from the coordination waters of calcium and carbonate ions that remain in the bulk mineral. Note that

the two environments do not form separate phases, i.e., there are no phase boundaries, at least not on a scale larger than several nm. This interpretation agrees remarkably well with previous investigations of ACC formation mechanisms²³ and offers a unifying explanation for the results from MAS NMR, C-AFM, TGA, and potentiometric titrations.

Fig. 5. Cartoon depicting the proposed formation mechanism of distinct water environments present in ACC (not to scale). Solute prenucleation clusters (PNC) undergo phase separation and form nanodroplets of a dense liquid phase (DLP) of the mineral. The polycarboxylate binds to calcium ions on their surface and colloidally stabilizes them. Eventually, the nanodroplets aggregate to reduce their interfacial free energy, and the species present on the surface of the droplets, as well as counter-ions present in the mother solution (mostly bicarbonate, and also hydroxide ions at pH > 9, shown in blue) are entrapped in the growing liquid-like mineral precursor via interface internalization (green areas). Due to the high polymer concentration and the different chemical environment on the surface of the DLP droplets, the distinct chemical environments will remain in the growing DLP and are transferred into solid ACC upon dehydration and solidification. As shown by MAS NMR, one of these environments is rigid and allows only restricted, anisotropic motion of the water molecules. We attribute this environment to the bulk of the original DLP nanodroplets. The other environment remains from imperfect coalescence of the DLP nanodroplets and hence consists of water molecules undergoing isotropic motion and kinetically entrapped hydroxide ions, which form a network across the mineral precursor.

Finally, as stated in the introductory part and repeated in other words in the final lines of the manuscript, in my view the statement that potentially designed new materials based on proton conductive amorphous minerals will be furthered is rather far-fetched.

Author Reply: We agree that practical applications of the conductive minerals will be realized far in the future and have removed the statement.

Revisions: Page 15, Conclusions section: Removed sentence and references regarding application examples

Ion-conducting materials are used as electrolytes,⁶⁸ ion-exchange membranes,⁶⁹ for example in fuel cells, or touch panels,⁷⁰ illustrating the tremendous potential of this class of green materials.

Overall, the SI is difficult to read by referring hence and forth, while if one consults the SI, one expects to be informed about relevant details at one spot. One typo is on page 4, second paragraph: "... sealed beaked ... " where beaker is meant, I suppose.

Author Reply: We thank the Reviewer for pointing this out and agree that the structuring of the SI needed to be improved. We have therefore reworked the supplementary information. Due

to the amount and complexity of the data, we, however, cannot eliminate every cross-reference within the SI, but we believe that that we have significantly improved readability.

Revisions: We have reworked the SI in a major revision:

- Order of SI Figures is now in order of appearance in the main manuscript (reference on supplementary images starts now from Fig. S1 in the main manuscript)
- additional discussion sections have been moved to the back of the SI
- added a list of supplementary images in the table of contents.

Supplementary Information, Page 5, line 14: Correction of typo

All experiments were performed in a sealed ~~beaked~~ beaker under N₂ shower, saturated with water vapor, to prevent diffusion of CO₂ into the solution.

On page 7 of the SI the authors indicate “samples were dried at 40 °C in vacuum overnight to remove surface adsorbed water”. Something similar is said on page 7 for storing overnight in the glovebox lock. I understand the low temperature (40 or room temperature) but I think, from experience, this is insufficient to remove adsorbed water.

Author Reply: We agree with the Reviewer that the presented procedure is not sufficient to remove surface adsorbed water quantitatively. However, this method was sufficient to lower the amount of surface adsorbed water to an extent that sample analysis was not negatively affected.

For MAS NMR and TGA studies, the drying procedure is crucial to remove any remaining solvents from synthesis (ethanol, acetone) and most of the surface adsorbed water. The presence of these solvents would cause measurement artefacts, while water would significantly reduce the stability of our ACCs in MAS NMR studies. ACCs are not stable when exposed to water, so crystallization would occur after a few days if they were not dried. Drying the ACCs with the presented method was sufficient to reduce the water to an extent where we did not have problems with ACC crystallization in case of polymer-stabilized ACCs.

For C-AFM studies, removal of surface bound waters is crucial to eliminate the pathway for conductivity through the surface water layer. Eliminating this pathway is especially important as this layer would be rich in ions due to the solubility of the amorphous minerals. We have demonstrated using vaterite reference samples that the procedure is sufficient to remove the water to an extent that no conductivity is detected. Also, no conductivity is detected if the ACC particles are only in loose contact to each other (see Fig. S13g-h); if there had been a surface layer of water, we would have expected to observe conductivity, which was not the case.

However, we agree with the Reviewer that these points need to be clarified in the manuscript. We have therefore removed the part stating the “removal of surface adsorbed water” and only referred to it as “drying” the samples. For C-AFM samples, however, we think it is crucial to emphasize that surface adsorbed waters must be removed to an extent that no conductivity is caused by a water layer on the surface. We have therefore added an additional sentence to the text.

Revisions: Supplementary Information, Page 9, line 7:

Measurements were performed with at least 10 mg of sample and prior to TGA analysis, samples were dried at 40 °C in vacuum overnight ~~to remove surface adsorbed water~~.

Supplementary Information, Page 21, line 2:

All measurements were performed in oxidative atmosphere (Ar:O₂ 80:20 v/v) and all samples were dried at 40 °C in vacuum prior to measurement ~~to remove surface adsorbed water.~~

Supplementary Information, Page 10, line 2:

Prior to C-AFM analysis, the samples were stored in the glovebox lock in vacuum overnight to remove surface adsorbed water, ~~as surface adsorbed water must be removed to an extent that no conductivity is caused by a water layer on the surface of the particles. Drying the samples in the glovebox lock proved to be sufficient for this, as shown by vaterite reference samples.~~

This brings me to an overall assessment from my side. In my opinion the experiments and interpretation have been done carefully, apart from the aspects mentioned. These aspects need attention. As there is no evident suggestion for a mechanism of inhibition, I leave the question whether the results are sufficient to warrant publication in Nature Communications to the editor.

Author Reply: We thank the reviewer for the positive comment on the experiments and interpretations. As mentioned before, we do provide the mechanism for mineralization inhibition as well as bicarbonate entrapment.

We are confident that the additional MAS NMR studies and the proposed model for ACC structure and formation complement the experimental data and altogether constitutes a comprehensive study on the mechanism of mineralization inhibition and ACC formation, including distinct water environments and an explanation for ACC conductivity (structure-property relations), thereby warranting publication in Nature Communications.

Reviewer #3:

The authors quantitatively analyzed the incorporation of bicarbonate into ACC in the presence/absence of polymer additives. I read this manuscript with great interest because I am also curious about why a small fraction of polyelectrolyte can effectively stabilize a relatively large amount of ACC. I was impressed by the authors' designs in the experiments, which provided solid evidence to support their conclusions. To further improve the quality of this work, the following issues should be considered.

Author Reply: We thank the Reviewer for the very positive feedback and their suggestions to improve the manuscript.

1. The conductivity is an important property of ACC as a material, and thus should be discussed more prudently. No evidence has been provided to reveal the phase of the large grain in Fig. 3c (SEM-Raman can be helpful). I also doubt if proton is the only species that is responsible for the conductivity; the current C-AFM tests remain semi-quantitative, and possibly further characterizations based on ACC samples with larger size and regular shapes could help.

Author Reply: We thank the Reviewer for this feedback and their suggestion. We have performed additional extensive MAS NMR studies on the ACC materials and have revised our interpretation of conductivity based on the new results. In hindsight, the reviewer was correct in doubting our explanation of conductivity. In our revised manuscript, we show that hydroxide ions are incorporated in the ACC, and that ACC possesses distinct water environments. In our revised interpretation, a high-dynamic water environment forms a network across the bulk of

ACC. In this high dynamic water-environment, water molecules show isotropic motion, and we propose that the hydroxide ions present in this environment cause the conductivity.

Regarding the large structure in Fig. 3c, we are convinced that this is the dried amorphous mineral precursor. We have detected many structures like the one presented in Fig 3c across the wafer. In fact, most of the precipitated material was present in these structures. We have closely monitored the phase of the mineral by ATR-FTIR, and we could not detect any signs of a crystalline mineral phase. Even minor amounts of crystalline CaCO_3 can easily be detected in FTIR, as the sharp crystalline ν_2 carbonate vibration at 872 cm^{-1} shows much higher intensity compared to the broad ν_2 carbonate vibrational band in amorphous CaCO_3 (864 cm^{-1}). If the investigated structures would contain even minor amounts of crystalline matter, this would have certainly been detected in FTIR.

2. There is essentially no direct comparison in the C-AFM tests between the ACC with/without bicarbonate incorporation. It can be seen from Fig. 2a that the temperature at which bicarbonate decomposes is lower than that for the crystallization of ACC. This allows the elimination of all/most of the bicarbonates in the sample while maintains the amorphous phase. Then it is possible to directly compare the conductivity of two samples: ACC with/without bicarbonate.

Author Reply: As is discussed in detail in the added MAS NMR section, we have revised our interpretation of ACC conductivity. Our results reveal that bicarbonate ions (or protons) do not cause conductivity, as we did not detect any dynamic bicarbonate species. Instead, we propose that kinetically entrapped hydroxide species, which we did detect in MAS NMR, are responsible for the conductivity. This also explains why conductivity is detected for polymer-free ACCs, which possess a much lower bicarbonate content, but are formed through the same mechanism (aggregation of dense liquid phase nanodroplets) and thus should exhibit similar structure-property relations, at least when it comes to the water environments and conductivity.

Considering the new results and importance of water environments in the ACC, we suggest being very careful with applying any type of significant heating to the ACC samples prior to analysis. Heating will affect (and at some point remove) the water in the ACC, which will significantly alter its (nanoscopic) structure. We propose that the water environments play a crucial role for ACC structure and its properties, and heating would actually remove these water environments.

3. In Fig. 2d, the small foot is assigned to bicarbonate. Yet the authors commented “However, its relative intensity is a few percent and it is therefore unlikely to be the bicarbonate species detected in the dense liquid phase and in the precipitate by TGA”. A contradiction is proposed but without proper discussion. It is also possible to obtain a hint by comparing the NMR of ACC with/without the incorporation of bicarbonate.

Author Reply: We agree with the reviewer that it would be nice to compare the experimental data with polymer-free, bicarbonate-free ACC. However, we did not manage to synthesize ACC in a way that it is stable enough for the MAS NMR experiments. As the NMR investigations take days up to sometimes weeks, our polymer-free ACCs always crystallized during the measurement. The problem of ACC crystallization induced by the centrifugal forces of MAS is well known, as described in Gebauer et. al., *Angew. Chem. Int. Ed.* **2010**, 49, 8889-8891.

The presented strategy for stabilizing the ACC (embedding the ACC particles in epoxy resin) was no option for us, and we always detected crystallization. Therefore, we focused on the polymer-stabilized ACCs in our studies, that did not show crystallization during measurement.

Please note, we have resolved the apparent contradiction with additional MAS NMR experiments and offer an explanation for the “missing” bicarbonate in MAS NMR.

Revisions: Page 10,11: A new section describes ^1H MAS NMR experiments and numerical simulations.

MAS NMR reveals two distinct water environments in polymer-stabilized ACC

The ^1H NMR spectrum of ACC stabilized by poly-aspartate at a spinning frequency of 10 kHz is shown in Figure 3a (black curve, analogous spectra at spinning frequencies of 5 and 0 kHz are shown in Fig. S11). A pattern of spinning sidebands is observed as well as a strong central peak with a broad base, which appears to push this pattern up. This agrees with observations by others for ACC of both synthetic and biological origin.^{46,47,50-52} A complete interpretation of the ^1H NMR spectra of ACC has, however, not been given.

By means of 2-dimensional WISE (wideline-separation) experiments (Fig. S14 and S15),⁵³ we obtained ^1H spectra of ACC, which are detected after a transfer of magnetization, via cross-polarization, to the ^{13}C nucleus of carbonate (purple curves in Fig. 3a, S11). Michel et al. already noted that for ACC these indirectly detected ^1H spectra differ from the directly detected ^1H spectra.⁴⁶ The strong central peak is absent and a Pake pattern, with spinning sidebands, due to the dipolar coupling between the two ^1H s of water is discernable.⁵⁴ This is characteristic of structural water in minerals with a low hydrogen density.⁵⁵

To assist in the interpretation of the ^1H spectra of ACC, we performed numerical simulations using the magnetic resonance simulation library Spinach.⁵⁶ We first simulated the indirectly detected ^1H spectra, taking the magnetic properties of monohydrocalcite (MHC) as a starting point. The crystal structure of MHC is known from X-ray diffraction and has also been investigated by quantum chemical methods.⁵⁷⁻⁶⁰ Simulations based on the calculated magnetic properties fit the experimental MAS NMR ^1H spectra of MHC remarkably well (Fig. S16). At room temperature, water molecules in many crystal hydrates undergo twofold rotations about the bisector axis (180° “flips”).⁶¹ If such flips are included in the simulation, the set of chemical shift anisotropies and dipolar couplings of MHC also produces a good simulation of the indirectly detected ^1H spectra of ACC (Fig. 3a, S11, blue curves). This strongly suggests that the ^{13}C -detected ^1H spectra arise from structural water molecules embedded in a rigid (but amorphous) environment of calcium carbonate, which allows only restricted, anisotropic motion.

Next, we turned to simulation of the directly detected ^1H spectra. The lack of structure (apart from a set of narrow lines, vide infra) and the width of the strong central peak (5-6 kHz) point to slow isotropic motion. To investigate this, we simulated the ^1H MAS NMR spectra of a water molecule undergoing rotational diffusion with varying correlation times using the stochastic Liouville equation, see Figure S17.^{62,63} For a correlation time of $1 \cdot 10^{-6}$ s, the simulated line width matches the width of the central peak in the experiments (green curves in Fig. 3a, S11). Taking the sum of the simulations of the indirectly detected ^1H spectra and the simulations of slow isotropic motion, here with equal weights, produces the red simulations in Figures 3a, S11, which reproduce all features of the experimental spectra (black curves).

The ^1H NMR spectra and the numerical simulations thus indicate the presence of a second, mobile environment in ACC, which allows water molecules to undergo isotropic motion. The correlation time of this motion is considerably longer than for bulk water, likely due to spatial confinement. As the temperature is lowered from 25 to -25°C (Fig. S18), the isotropic motion slows down further, causing the central peak to come down, while the broad base comes up.

We previously noted that the magnetization transfer by cross-polarization is inefficient for the ^{13}C nuclei of carbonate in ACC (Fig. 2a). From the WISE experiments, we now know that the structural water molecules, which undergo only restricted, anisotropic motion, are the source of this magnetization. Numerical simulations in Figure S19 show that the 180° flips we invoked to simulate the indirectly detected ^1H spectra of ACC can also account for an inefficient transfer in a cross-polarization experiment. ^1H - ^{13}C cross polarization spectra recorded at slow spinning frequency (2 kHz, Fig. S20) reveal the chemical shift anisotropy (of the ^{13}C nuclei of carbonate in ACC). The observed pattern of spinning sidebands does not change as the temperature is decreased to -25°C and is the same as recently observed for carbonate in a frozen carbonate solution (pH 7) at 100 K.⁶⁴ Thus, carbonate itself does not undergo motion, or only on very slow

time scales. Furthermore, directly detected ^{43}Ca MAS NMR spectra show only broad, featureless resonances, hence giving no indication of mobility of calcium ions.⁶⁵ These observations together confirm that the rigid environment consists of amorphous calcium carbonate with embedded structural water molecules, which only undergo restricted, anisotropic motion, and 1-2% of bicarbonate.

Figure 3b shows the central region of the directly detected ^1H NMR spectrum of ACC in detail. A set of narrow lines (widths in the range 15-50 Hz) is superimposed on the strong central peak (width 5-6 kHz). We assign these lines to hydroxide ions in rapid exchange with the water molecules that make up the second, mobile environment.⁴⁷ Possibly the different observed chemical shifts relate to local variations in basicity. The presence of hydroxide ions within the mobile water environment can make ACC conductive. We explored this experimentally.

Fig. 3. a) Directly and ^{13}C -carbonate detected ^1H MAS NMR spectra of PAsp-stabilized, 10% ^{13}C -carbonate ACC (PAsp_disACC), with accompanying simulations, at a spinning frequency of 10 kHz and room temperature. The ^1H s from PAsp do not significantly alter the shape of the directly detected spectra, see Figure S12. The MAS NMR probe, however, gives rise to a broad ^1H NMR background signal, which has been removed from the directly detected ^1H spectrum (black) following a procedure outlined in the supporting information (Fig. S13). b) Central region of the directly detected ^1H NMR spectrum of 100% ^{13}C -carbonate ACC stabilized by PAsp (PAsp_disACC), at a spinning frequency of 10 kHz and room temperature.

Revisions: Page 7, line 30: Added discussions on bicarbonate content

Weight loss observed by TGA-MS-IR (~13%, corresponding to roughly 27% of bicarbonate present in the ACC, see Fig. S9e) agrees with the amount of bicarbonate kinetically entrapped within the DLP as determined in the titration experiments (Fig. 1d). Yet, direct investigation of intact ACC nanoparticles by MAS NMR indicates a much lower bicarbonate content. To accommodate this apparent discrepancy, we propose the following reactive processes.

- (i) Upon dehydration and solidification, DLP-entrapped bicarbonate ions react with OH^- ions that are abundant in the dense and lean aqueous phases at basic conditions. While the OH^- /bicarbonate population is equilibrated in the liquid state, without hydration water, the polarizing power of Ca^{2+} destabilizes the bicarbonate ion in the solid state. Consequently, no solid calcium bicarbonates are known to occur at ambient conditions, and 1-5% bicarbonate may be the maximum amount that can be accommodated within solid ACC.²⁰ Hence, upon dehydration and solidification of the DLP, the following process occurs: $\text{DLP}\cdot\text{HCO}_3^- + \text{OH}^- \rightarrow \text{ACC}\cdot\text{CO}_3^{2-}\cdot\text{H}_2\text{O}$ (note that ACC usually contains an excess of around 1 mole of water per mole of calcium carbonate).
- (ii) The kinetic entrapment of bicarbonate ions causes the DLP to be locally calcium-ion-deficient with respect to carbonate, as two bicarbonates are expected to balance the charge of one calcium ion within the macroscopically neutral DLP. This imbalance is transferred into solid ACC by process (i). Local

structural water molecules stabilize these sites, which we label as $\text{Ca}^{(-)}$. Upon heating in TGA and removal of stabilizing water molecules, carbonate ions close to calcium-deficient sites then decompose according to the following process: $\text{Ca}^{(-)}\text{CO}_3 \cdot \text{H}_2\text{O} \rightarrow [\text{Ca}^{(-)}\text{HCO}_3]^{+} + \text{OH}^{-} \rightarrow \text{Ca}^{(-)}(\text{OH})_2 + \text{CO}_2 \uparrow$

Indeed, we can detect a corresponding calcium hydroxide fraction in XRD measurements of the samples after TGA analysis (Fig. S10). The above processes lead to calcium carbonate decomposition via calcium bicarbonate at correspondingly low temperatures in TGA. Quantitatively, this corresponds to the extent of kinetic bicarbonate entrapment in the DLP, even though not being present as bicarbonate in solid ACC at ambient conditions. Rather, the polycarboxylate-mediated bicarbonate entrapment in the DLP leads to locally calcium-deficient solid ACC, which can markedly contribute to the stabilization against nucleation of a solid. Thus, we observe a direct correlation between these effects as discussed above. This, however, begs the question how the charges are balanced within the ACC phase, globally, as phases are certainly macroscopically neutral. Sodium ions do not seem to play a major role: The exothermic decomposition of bicarbonate species in polymer-stabilized ACC (Fig. 2b) differs fundamentally from that of NaHCO_3 , which exhibits endothermic decomposition (Fig. S5d). In addition, no sodium-containing phase was detected in the XRD spectra of the product isolated after TGA (Fig. S10), further confirming that no significant NaHCO_3 coprecipitation took place upon the quenching and isolation of the ACC. It is possible that some sodium ions are incorporated within the ACC for local charge balancing, but it rather appears that negative charges at calcium-deficient sites may be globally balanced by excess calcium binding within the phase in other environments, likely in proximity of the incorporated polymer. In any case, all of the above implies that also the water sites in the polymer-stabilized ACC can play a crucial role, and these were further studied by MAS NMR.

4. "In polymer-free experiments, carbonate ions and calcium ions are bound in PNCs, resulting in the formal removal of carbonate ions from the buffer equilibrium ($\text{HCO}_3^{-} \rightleftharpoons \text{H}^{+} + \text{CO}_3^{2-}$) and release of H^{+} ions that are then neutralized by automatic NaOH addition... Thus, from the NaOH addition, the removal of carbonate species from the buffer equilibrium upon addition of calcium due to ion association can be calculated." I found these sentences misleading that I was thinking the authors calculated the amount of carbonate ions simply based on this equation: $n(\text{NaOH}) = n(\text{H}^{+} \text{ ion release}) = n(\text{carbonate bound by Ca})$. This equation is obviously not correct because the equilibrium shifts as the total amount of carbonate species (carbonate and bicarbonate) decreases. It seems the authors have already taken this into account (SI, 2.3.3 and 2.3.4). Anyway, I suggest that these sentences should be rephrased.

Author Reply: We indeed have taken this into account, as described in the sections in the SI that the reviewer has correctly referred to. We have added an additional sentence in the manuscript to clarify that the shift in the buffer equilibrium was considered in the calculations.

Revisions: Page 4, line 4: Further clarification was added:

In polymer-free experiments, carbonate ions and calcium ions are bound in PNCs, resulting in the formal removal of carbonate ions from the buffer equilibrium ($\text{HCO}_3^{-} \rightleftharpoons \text{H}^{+} + \text{CO}_3^{2-}$). Protons are formed due to the regeneration of carbonate from bicarbonate ions upon re-equilibration at constant pH. In the titration experiment, these protons are then neutralized by automatic NaOH addition to keep the pH constant. Therefore, the extent of NaOH addition in the prenucleation regime can be tracked to assess the carbonate binding (Fig. 1b), confirming a calcium : carbonate ion binding ratio for polymer-free reference experiments of 1:1 (Fig. 1c).

5. Fig. 1b has not been properly cited.

Author Reply: We thank the reviewer for pointing this out and have added the missing reference to Fig. 1b.

Revisions: Page 4, line 9: Added a sentence to properly reference Fig. 1b:

Therefore, the extent of NaOH addition in the prenucleation regime can be tracked to assess the carbonate binding (Fig. 1b),[...]

6. Inconsistency of the abbreviations throughout the manuscript.

Author Reply: We have checked the manuscript for inconsistencies and have indeed detected a few instances, at which Pasp was used instead of PAsp. We have corrected these typos and carefully checked the manuscript for more inconsistencies.

Revisions: Replaced Pasp by PAsp in several instances.

7. The scale factor maximized at pH = 9.0. What is the rationale that the “specific pH value” is set to 9.8?

Author Reply: In the original conception of our work and NMR studies, we put particular emphasis on the role of ACC proto-structure in polymer-controlled crystallization processes (see Gebauer et. al., *Angew. Chem. Int. Ed.* **2010**, 49, 8889-8891 for more information on ACC proto-structures). We therefore started our NMR work using a pH value of 9.8, which is the pH value leading to proto-vaterite ACC. In addition, a larger amount of ACC can be isolated if the synthesis is performed at higher pH values in presence of polymers (due to the higher fraction of carbonate ions in the solution), so pH 9.8 was used as a starting point.

In the course of the study, we saw that the bicarbonate content incorporated in the ACCs plays an important role, and we shifted our focus from investigating the role of ACC proto-structure to investigating the role of bicarbonate. Due to the numerous experiments already performed at pH 9.8 and the well-established ACC synthesis procedure, we decided to keep this pH value as primary pH value for our experiments. In hindsight, perhaps investigations at pH 9.0 would have been preferable considering the maximized inhibition capability, however, a detailed analysis of the observed pH effect is beyond the scope of the present study. As the bicarbonate fraction is still major at pH 9.8 and the basic mechanisms of ACC formation (apart from proto-structure) do not seem to be significantly affected by pH, at least according to the current state of the art, we expect that the outcome would have been very similar at pH 9.0. However, certainly, future work would be warranted to check if this is really the case or not.

8. The vaterite nanoparticles, which were not stable in water, were washed with deionized water. The authors should provide the evidence to show its phase (instead of “data not shown”).

Author Reply: We did analyze the preparation using ATR-FTIR spectroscopy, which showed a typical spectrum of vaterite. We decided to not include the analysis, as the synthesis of vaterite is literature-known, we did check that it was successful, and the sample is merely a reference sample for AFM experiments. For the same reasons, we also did not present the TEM analysis of the other C-AFM reference sample (Au nanoparticles).

As the SI is already extensive, we would prefer to not add this data, especially since issues with readability have been pointed out by the Reviewers. However, if the Reviewer insists on the importance of the data, we will of course be happy to include it.

Below, we show the ATR-FTIR spectrum of the vaterite nanoparticles used as C-AFM reference particles.

Figure: ATR-FTIR characterization of vaterite reference particles for AFM analysis. The vibrational bands at 872 and 844 cm^{-1} confirm the presence of vaterite. No calcite (band at 712 cm^{-1}) was detected.

9. The SI file is terribly formatted.

Author Reply: As mentioned in our reply to Reviewer 2, we agree that the structure of the SI needed to be improved. We have therefore reworked the supplementary information, and hope that we could significantly improve the readability.

Revisions: We have reworked the SI in a major revision:

- Order of SI Figures is now in order of appearance in the main manuscript (reference on supplementary images starts now from Fig. S1 in the main manuscript)
- additional discussion sections have been moved to the back of the SI
- added a list of supplementary images in the table of contents.

Reviewers' Comments:

Reviewer #1:

Remarks to the Author:

The quality of the data is technically sound; they have been interpreted carefully, and presented in sufficient detail as well.

The conclusions have been properly included and advance the understanding in a way that will move the field forward.

In any case, I would like to confirm my original report of Referee.

Reviewer #2:

Remarks to the Author:

The authors did some additional experiments, responded in detail to the issues raised and revised the manuscript accordingly, apart from several small modification that are also highlighted. I will respond in the order of the issues raised in my first review (2nd reviewer).

1) removal: ok.

2) TGA below 100 C: The authors appropriately added a reference for the water. However, I am not convinced that ethanol can be removed at 40 C and in the absence of mass spectroscopy data, this remains unclear. Hence, I would appreciate if the something like this is said.

3) 12 C: clear and agreed.

4) unexplained X-ray peaks: I understand this, but I think the authors just should say that clearly, the more so because the peaks are not weak.

5) shape anisotropy: clear, but a one or two lines including the references would help readers a great deal.

6) grain boundaries: clear and agreed.

7) conductivity: clear.

8) two environments: see item 13.

9) potential use: ok

10) SI: I understand that for a manuscript like this one, no cross-referencing is rather difficult. The revised version is much better.

11) typo: ok.

12) drying at 40 C: clear and agreed.

13) two mechanisms: with the addition given, the authors have clarified this a great deal.

In conclusion, in my opinion the manuscript can be published after a few sentences on the remaining issues are added. As the authors and I essentially agree, there is no need for me to see the manuscript again after these sentences are added.

Reviewer #3:

Remarks to the Author:

I appreciate the improvements made by the authors especially in Figure 5 and the SI. I read their responses to all the issues and I think they have addressed them properly.

Point-by-point Replies (NCOMMS-23-05524)

Herein, our replies appear indented and italicized, corresponding changes are highlighted in yellow.

Reviewer #1 (Remarks to the Author):

The quality of the data is technically sound; they have been interpreted carefully, and presented in sufficient detail as well. The conclusions have been properly included and advance the understanding in a way that will move the field forward.

In any case, I would like to confirm my original report of Referee.

Author Reply: We thank the reviewer for the positive comments on our data and manuscript.

Reviewer #2 (Remarks to the Author):

The authors did some additional experiments, responded in detail to the issues raised and revised the manuscript accordingly, apart from several small modification that are also highlighted. I will respond in the order of the issues raised in my first review (2nd reviewer).

Author Reply: We thank the reviewer for the feedback, and address the remaining comments below.

1) removal: ok.

2) TGA below 100 C: The authors appropriately added a reference for the water. However, I am not convinced that ethanol can be removed at 40 C and in the absence of mass spectroscopy data, this remains unclear. Hence, I would appreciate if the something like this is said.

Author Reply: We are happy to elaborate further on this topic. Indeed, we claim that the mass loss below 100 °C cannot be attributed to ethanol and we do have data in the manuscript that supports this interpretation. On the one hand, no ethanol signals are visible in the NMR spectra of the investigated samples (see Fig. 2a in the main manuscript). In addition, we provide TGA-IR data for the investigated samples (see Fig. 2e in the main manuscript). In case of significant amounts of ethanol contained in the sample, this ethanol release would be visible in the IR spectra, i.e., C-H stretching vibrational bands around 3000 cm⁻¹ would be visible. As evident by the TGA-IR spectra (Fig. 2e), these bands are not visible below 100 °C and only water vapor was detected. We therefore are confident that the main reason for the significant decrease in weight below 100 °C can be attributed to water. Therefore, ethanol can only be contained in negligible amounts in the samples that are undetectable by the employed NMR and TGA-MS-IR techniques. We agree that this is an important point to mention and have added this explanation in the SI for further clarification.

Revisions: Page 7 in the SI, additional clarification added:

It needs to be emphasized that although ethanol and acetone were used in the ACC synthesis, these solvents are not present in the dried and investigated samples. No significant amounts of solvents were detected in NMR experiments (see Fig. 2a in the main manuscript) and no vibrational bands of organic solvents were detected in TGA-IR measurements (see Fig. 2e in the main manuscript). Therefore, the weight loss below 200 °C can be attributed to the release of water.

3) 12 C: clear and agreed.

4) unexplained X-ray peaks: I understand this, but I think the authors just should say that clearly, the more so because the peaks are not weak.

Author Reply: We have added our response the reviewer's comment in the caption of the image in the SI. Thereby, it is clearly stated to the reader that we were unable to identify the peaks and which database we have used.

Revisions: Page 14 in the SI, additional clarification added:

The available XRD database (Cambridge Structural Database) was checked to identify these reflections, but no assignment was possible. These reflections were also compared to those of all listed sodium containing compounds (especially oxides that might be formed in TGA) but no compound could be identified.

5) shape anisotropy: clear, but a one or two lines including the references would help readers a great deal.

Author Reply: We have added our response the reviewer's comment in the caption of the image in the SI, including the references described in our previous reply.

Revisions: Page 27 in the SI, additional clarification added:

In addition, the samples were prepared using spin-coating, which exerts additional forces on the particles. Similar effects of shape-anisotropy of sedimented amorphous calcium carbonate particles were described in earlier studies.^{21,22}

6) grain boundaries: clear and agreed.

7) conductivity: clear.

8) two environments: see item 13.

9) potential use: ok

10) SI: I understand that for a manuscript like this one, no cross-referencing is rather difficult. The revised version is much better.

11) typo: ok.

12) drying at 40 C: clear and agreed.

13) two mechanisms: with the addition given, the authors have clarified this a great deal.

In conclusion, in my opinion the manuscript can be published after a few sentences on the remaining issues are added. As the authors and I essentially agree, there is no need for me to see the manuscript again after these sentences are added.

Author Reply: We have included the suggested clarifications and once again thank the reviewer for the comments that have significantly improved the manuscript.

Reviewer #3 (Remarks to the Author):

I appreciate the improvements made by the authors especially in Figure 5 and the SI. I read their responses to all the issues and I think they have addressed them properly.

Author Reply: We are glad that we have addressed the reviewer's suggestions properly and thank the reviewer for their comments that have significantly improved the manuscript and the SI.